# Hypoxia Imaging in Lung Cancer: A PET-Based Narrative Review for Clinicians and Researchers

**DOI:** 10.3390/ph18040459

**Published:** 2025-03-25

**Authors:** Ahmad Alenezi, Hamad Alhamad, Aishah Alenezi, Muhammad Umar Khan

**Affiliations:** 1Radiologic Sciences Department, Kuwait University, Kuwait City 31470, Kuwait; 2Occupational Therapy Department, Kuwait University, Jabriya 31470, Kuwait; 3Nuclear Medicine Department, Jahra Hospital, Ministry of Health, Al Jahra 03200, Kuwait

**Keywords:** hypoxia, positron, emission, tomography, cancer

## Abstract

**Background:** Hypoxia plays a critical role in lung cancer progression and treatment resistance by contributing to aggressive tumor behavior and poor therapeutic response. Molecular imaging, particularly positron emission tomography (PET), has become an essential tool for noninvasive hypoxia detection, providing valuable insights into tumor biology and aiding in personalized treatment strategies. **Objective:** This narrative review explores recent advancements in PET imaging for detecting hypoxia in lung cancer, with a focus on the development, characteristics, and clinical applications of various radiotracers. **Findings:** Numerous PET-based hypoxia radiotracers have been investigated, each with distinct pharmacokinetics and imaging capabilities. Established tracers such as ^18^F-Fluoromisonidazole (^18^F-FMISO) remain widely used, while newer alternatives like ^18^F-Fluoroazomycin Arabinoside (^18^F-FAZA) and ^18^F-Flortanidazole (^18^F-HX4) demonstrate improved clearance and image contrast. Additionally, ^64^Cu-ATSM has gained attention for its rapid tumor uptake and hypoxia selectivity. The integration of PET with hybrid imaging modalities, such as PET/CT and PET/MRI, enhances the spatial resolution and functional interpretation, making hypoxia imaging a promising approach for guiding radiotherapy, chemotherapy, and targeted therapies. **Conclusions:** PET imaging of hypoxia offers significant potential in lung cancer diagnosis, treatment planning, and therapeutic response assessment. However, challenges remain, including tracer specificity, quantification variability, and standardization of imaging protocols. Future research should focus on developing next-generation radiotracers with enhanced specificity, optimizing imaging methodologies, and leveraging multimodal approaches to improve clinical utility and patient outcomes.

## 1. Introduction

Most recent statistics (as of 2022) indicate that lung cancer (LC) remains the most prevalent type of cancer, constituting nearly 11% of all cancer cases across genders [1]. The two primary histological types of lung cancer are non-small-cell lung cancer (NSCLC) and small-cell lung cancer (SCLC) [2]. NSCLC encompasses several histological subtypes, with adenocarcinoma emerging as the most common subtype [2].

Likewise, in recent years, the field of molecular imaging has transformed oncological imaging by providing noninvasive evaluation of tumor pathophysiology. This technology provides a comprehensive characterization of the state of disease, prognosis, and therapeutic response, making it an essential tool in the fight against cancer [3]. Nuclear medicine is a pivotal field which leverages molecular imaging techniques, utilizing radionuclides to diagnose and treat various medical conditions. Radiopharmaceuticals, key agents in this domain, facilitate imaging modalities like single photon emission tomography (SPECT) and positron emission tomography (PET) [4], enabling noninvasive visualization of specific biological processes through the accumulation and kinetics of radiotracers [3], thereby making nuclear medicine an integral tool in both the imaging and treatment of many oncological diseases, including LC.

PET is advanced imaging technology which has the potential to visualize various tracers that target the altered biological processes observed in malignant tumors [5]. To develop effective diagnostic tracers for PET imaging, researchers used cancer hallmarks as a conceptual tool to describe the various mechanisms of cancer biology [5,6,7]. PET’s ability to visualize some of these altered states in cancer cells has also helped to improve our understanding of tumor biology and heterogeneity, thereby facilitating the development of more targeted and effective treatments for cancer [8]. Importantly, PET imaging has both diagnostic and prognostic importance and is commonly used for various types of malignancies for staging the disease, treatment evaluation and follow-up.

An ideal radionuclide for PET imaging is one that has an adequate half-life for imaging and a short positron range. This results in high-quality imaging outcomes and accurate diseased tissue localization [9]. Depending on the purpose of the imaging process, specific ligands are carefully chosen to visualize factors such as glucose metabolism, cell proliferation, somatostatin receptor expression, hypoxia, or angiogenesis. It is critical that these ligands exhibit high specificity and sensitivity in detecting particular types of cancer. This ensures that accurate and reliable imaging results aid the diagnosis and treatment of lung cancer patients.

Although there have been significant improvements in LC treatment in recent years, the presence of tumor hypoxia and other micro-environmental factors still limits the ability to achieve desirable treatment outcomes. Hypoxia is caused by inadequate blood flow and a subsequent low oxygen concentration [O_2_] and pO_2_ tension, which can lead to resistance to conventional cancer treatments, increased tumor aggressiveness, and unfavorable prognoses. The hypoxic condition arises from the rapid growth of aggressive cancer cells which consume a large amount of oxygen and might limit blood circulation. Under hypoxic conditions, cancer cells have a greater tendency to metastasize or initiate various types of resistance such as apoptosis resistance, genomic instability, or increased resistance to radiation and chemotherapy. Hypoxic cells are two or three times more resistant to radiation than normoxic cells [10,11]. This requires an increase in the radiation dose to kill malignant cells and prevent tumor relapse effectively. However, it has been found that increasing the radiation dose may exceed standard tissue tolerance and lead to unfavorable consequences [12,13,14]. Limiting the maximum dose of radiation in the chest area is important to prevent toxicity to normal tissue from radiation exposure. Due to all of these reasons, treating lung malignancy seems complicated and may result in lower patient survival rates.

In recent years, numerous studies have been conducted to provide an overview of the molecular basis of hypoxia in different physiological and pathophysiological conditions. The gold standard for evaluating tumor hypoxia is measuring the tissue oxygen pressure. This method involves the invasive measurement of tissue oxygen levels using a polarographic Eppendorf electrode. Höckel and Vaupel carried out the first experiments in the early 1990s [10,11]. However, the Eppendorf electrode has several disadvantages which cause tissue degradation and allow only one measurement per territory. Furthermore, this method involves invasive procedures and is only suitable for tumors which are easily accessible. Another limitation of this technique is that it cannot differentiate between necrotic and viable anoxic tissues. Consequently, it is not practical for regular clinical practice.

PET is considered one of the novel noninvasive imaging tools used to understand the molecular basis of hypoxia. Hypoxia PET imaging involves using nitroimidazole tracers or [Cu]-^64^Cu-ATSM. Recently, it has been suggested that ^89^Zr-girentuximab could indirectly assess variations in the intracellular pH due to hypoxia.

In this article, we aim to discuss the various aspects of PET imaging related to hypoxia radiotracers, particularly for lung malignancy, and propose future directions for better hypoxia detection and radiopharmaceutical design planning. Understanding the pathophysiology of hypoxia, its consequences, and the associated signaling pathways seems essential. Therefore, we will briefly highlight and explain these aspects as well.

## 2. Pathophysiology of Hypoxia

Hypoxia is a pathological condition caused by an oxygen supply which is inadequate for cellular metabolism in living tissues or the entire body. It occurs in approximately 50–60% of solid tumors, with significant variability between individuals and tumor types [15,16,17,18]. Typically, tumor hypoxia may occur due to poor perfusion, oxygen diffusion deficiency, or a low hemoglobin level associated with anemia. Hypoxic tumors exhibit morphological and functional abnormalities in newly formed capillaries caused by the imbalance between oxygen consumption and delivery [19]. Acute hypoxia is a type of hypoxia which occurs in tumors due to abnormal perfusion, which leads to a lack of oxygen delivery. On the contrary, chronic hypoxia is caused by an abnormality in oxygen diffusion [20].

Lack of oxygen (O_2_) is a serious health issue which can cause chronic diseases and even death. This is because O_2_ plays a significant role in nourishing cells and maintaining normal cellular and subcellular processes and mechanisms [21,22,23]. A deficiency in the pivot element has been associated with several severe illnesses, like resistant malignancy. The level of partial oxygen pressure (pO_2_) required for proper cellular function varies in different tissues. The threshold level below which cellular function is impacted negatively depends on the normal pO_2_ of the corresponding tissue. To maintain cellular oxygen homeostasis, there should be an equilibrium between oxygen delivery and demand [23].

Generally, malignant cells use a significant amount of oxygen, which will ultimately lead to failure in the oxygen equilibrium and result in hypoxia. The hypoxic conditions may trigger cancer cells to kick off angiogenesis, which is the process of developing new blood vessels from the already existing ones. Various studies support the positive correlation between hypoxia and angiogenesis processes in tumors. This relationship may cause an increase in blood flow to provide the tumor with the necessary oxygen and nutrients to grow, which can lead to metastasis [23]. Furthermore, chronic hypoxia may lead to corruption in the microenvironment which gives rise to variation in the chemokine, cytokine, growth factor, and reactive oxygen species (ROS) equilibrium. This imbalance, in turn, leads to deregulation in many protein complexes responsible for DNA transcription (e.g., NF-kB and HIF-1). This deregulation may, therefore, allow suitable survival for the de novo growth of cancer as well as its corresponding metastatic processes and other autoimmune complications [24,25,26]. This important connection between hypoxic cellular pO_2_ and disturbance in many molecular processes creates the need to study hypoxia more thoroughly and carefully rather than following a single molecular process. In fact, hypoxia should be distinguished from anoxia and necrosis (See Figure 1). This can be achieved by carefully estimating the cellular pO_2_ levels in contrast with the vascular pO_2_ levels using suitable imaging and gauging devices. Figure 1 classifies the biomolecular processes used in lung cancer hypoxia detection.

## 3. Narrative Survey

A broad literature search was conducted to explore the use of PET for assessing hypoxia in cancer. Relevant studies published between 1990 and 2025 were considered, with a focus on original research and comprehensive reviews discussing imaging modalities for hypoxia detection. Studies which did not specifically address hypoxia imaging for cancer, lacked methodological detail, or were limited to commentaries, editorials, or conference abstracts without full-text availability were not included in the discussion.

The literature search identified numerous studies (approximately 730 articles) related to PET, PET/CT, and PET/MR imaging. Articles were selected based on their relevance to the topic, emphasizing key findings and advancements in hypoxia imaging. The selection process was guided by expertise in nuclear medicine, radiopharmacy, and imaging sciences, ensuring a well-rounded perspective. Key studies which contributed significant insights into hypoxia detection were highlighted and discussed in the review.

A total of 189 research references related to PET-based hypoxia detection and imaging were analyzed for this narrative review. Studies were included if they focused on PET imaging of tumor hypoxia, discussed the use of radiopharmaceuticals for hypoxia detection, or provided insights into the clinical and preclinical applications of hypoxia imaging in oncology. Studies were excluded if they lacked a direct focus on PET-based hypoxia imaging, were limited to commentary or editorial pieces without original data, or primarily discussed imaging modalities unrelated to PET.

Among the analyzed studies, 50% (n = 95) addressed tumor hypoxia in a general context, covering aspects such as molecular mechanisms, imaging techniques, and the impact of hypoxia on cancer progression and treatment resistance. A substantial portion (about 39% (n = 75)) specifically examined tumor hypoxia in lung malignancies, including non-small-cell lung cancer (NSCLC) and small-cell lung cancer, emphasizing its prognostic significance and role in treatment outcomes. Additionally, 8% (n = 15) explored hypoxia in other cancer types, such as head and neck cancers, breast cancer, and gliomas. A small fraction of the studies (2% (n = 4)) was identified as unrelated to tumor hypoxia, and they were excluded from further discussion.

Through this review, we identified nine radiopharmaceuticals commonly used to evaluate tumor hypoxia, which are summarized in Table 1.

### 3.1. Hypoxia as a Limiting Factor for Cancer Therapeutics

In 1912, Swartz was the first person to observe a correlation between blood flow and radiation damage. He discovered that the skin’s reaction to radium decreases if the applicator presses hard on the skin, and this is attributed to a reduction in the blood supply [27]. However, Petry believed that there was a more complex mechanism to describe these findings. In fact, during his experiments on vegetable seeds, he was the first scientist to prove a correlation between oxygen levels and radiosensitivity [28,29]. Following these findings, Thomlinson and Grey conducted a pilot histopathological study in 1955, which revealed a correlation between oxygen and treatment effectiveness. They demonstrated that the effectiveness of radiotherapy is enhanced by oxygenation, while hypoxia diminishes it [30]. Since then, hypoxia has emerged as a popular research area for many leading cancer research institutions.

Moreover, it has been reported in the last two decades that a higher amount of hypoxia results in a worse prognosis during cancer treatment. In simple terms, the chances of successful treatment increase when the cellular microenvironment is at normoxic or near-normoxic levels [31]. In contrast with normoxic cells, hypoxic cells experience inefficient molecular signaling pathways due to a change in tissue histology, which primarily hinders the corrective processes (i.e., pO_2_ level restoration) and causes morphological changes leading to de novo tumor vasculature [32,33]. Aside from this, a low vascular density often fails to provide enough oxygenation to the affected tissue and gives rise to temporal and spatial variations during tumor adaptation to the pO_2_ levels [34] (see Figure 2). This variation in pO_2_ levels results in minimizing chemotherapeutic and radiotherapeutic effects through a variety of mechanisms (e.g., physiological and genomic mechanisms) (see Figure 3).

### 3.2. Molecular Mechanisms of Cancer Resistance to Radiotherapy and Chemotherapy

Radiation therapy (RT) is a common treatment for solid tumors, often used alongside chemotherapy, immunotherapy, or surgery. RT damages DNA through ionizing radiation, causing direct damage when radiation is absorbed by the DNA itself and indirect damage through radiolysis of the surrounding water molecules, leading to the formation of reactive oxygen species (ROS) [11,14,38]. The presence of oxygen enhances ROS formation, increasing cellular damage, while hypoxia reduces this effect, leading to radiotherapy resistance [39]. Tumor cells attempt to repair radiation-induced DNA damage through mismatch repair, base excision repair, nucleotide excision repair, and double-strand break (DSB) repair, primarily via non-homologous end joining (NHEJ) and homologous recombination [14,39]. However, due to dysfunctional DNA repair and hypoxia, cancer cells may survive radiation treatment, reducing its effectiveness [14,39].

Hypoxia also influences gene expression, particularly through hypoxia-inducible factor (HIF-1α), which upregulates the genes involved in angiogenesis and apoptosis (e.g., VEGF-1 and p53) [40,41]. Additionally, HIF-1α is associated with chemotherapy resistance, further complicating cancer treatment [31]. Given that hypoxia is a negative prognostic marker, precise and efficient detection methods are crucial. Traditional monitoring techniques, such as oxygen polarographic needle electrodes and immunohistochemical analysis, are invasive [31,42]. PET imaging has emerged as a promising noninvasive tool for real-time hypoxia monitoring through radiolabeling specific targets and receptors, allowing visualization of tumor metabolism and oxygenation levels in vivo [43]. This highlights the critical role of PET imaging in addressing hypoxia-related treatment challenges and improving therapeutic outcomes.

### 3.3. Hypoxia PET Imaging

Both nuclear medicine modalities, including PET and SPECT, use gamma ray photons to create an image [44]. SPECT imaging involves the use of hypoxia-specific compounds combined with gamma-emitting radioisotopes like ^123^I/^125^I [45,46] and ^99m^Tc [47] to show hypoxic areas within tumors. Although SPECT imaging is easy to perform, PET is generally preferred for its higher specificity in detecting hypoxic tissue. PET imaging relies on administering targeted drugs labeled with radioactive isotopes called tracers [43,48]. The term used to refer to a drug which has been labeled with a radioactive tracer is radiopharmaceutical. After administering the radiopharmaceutical, it emits positrons. Each positron travels a few millimeters before colliding with an electron in the surrounding tissue [49]. The collision of the two particles results in the emission of two photons, which can be detected and used to produce images that depict the particles’ origin and distribution [50]. Various tracers have been developed to detect hypoxic regions in tumors, each with unique advantages and use in PET imaging [51].

#### 3.3.1. ^18^F-Fluromisonidazole (^18^F-FMISO)

Selectively binding to hypoxic cells in vitro and in vivo is ^18^F-FMISO [17,52]. It is hydrophilic and can passively diffuse through cell membranes in normal tissues. Moreover, it diffuses and selectively binds to intracellular macromolecules after reduction with nitro-reductase enzymes in the cytoplasm within viable regions (see Figure 4) [53]. This binding is reversible when cells are well oxygenated. On the contrary, under hypoxic conditions, ^18^F-FMISO is gradually reduced, producing hydroxylamine (R-NHOH) compounds which covalently bind to intracellular proteins, resulting in metabolic trapping of the radiotracer (see Figure 4 and Figure 5) [54]. Importantly, ^18^F-FMISO is characterized by its slow reaction mechanisms and washout (2–4 h) from the normoxic cells as it regenerates after reoxidation [53]. Subsequently, using ^18^F-FMISO requires prolonged examination protocols to identify and quantify hypoxic tumor areas [53]. Therefore, the timing of passive diffusion, intracellular reduction, and trapping is vital for achieving good image contrast [55].

For assessing tumor hypoxia in patients with lung, brain, and head and neck cancer, ^18^F-FMISO has successfully been used [17,52,56,57,58,59]. Additionally, it has been used in benign medical conditions (e.g., to assess ischemia in patients with suspected coronary artery obstruction) [60]. Several clinical studies have investigated the potential role of ^18^F-FMISO in evaluating different aspects of head and neck cancer (HNC). These studies have evaluated various aspects of HNC, including correlation with other hypoxia biomarkers during radio-chemotherapy [17,61,62], its ability to predict outcomes in patients with HNC [63], and its uptake changes during serial imaging [16,64].

A clinical study was conducted on carbon ion radiotherapy patients with NSCLC which showed significant uptake of ^18^F-FMISO before radiotherapy and a decrease post therapy [65]. Furthermore, the association of ^18^F-FMISO uptake with the glioma tumor grade, hypoxia biomarkers (CA-IX and HIF-1α), and angiogenesis markers (VEGF) has also been reported [66]. The same study found a reduction in ^18^F-FMISO accumulation following chemotherapy, but the reason for this was not known. Some evidence suggests that in cases of severe hypoxia, where the level of oxygen (pO_2_) falls below 2–3 mmHg, the uptake of ^18^F-FMISO may not accurately indicate the level of hypoxia. This could be due to either a threshold point beyond which tracer uptake in the cell may not increase, or there may be a bioreduction in ^18^F-FMISO which does not exceed a certain point [67]. One study indicated that the use of ^18^F-FMISO may cause harmful side effects such as peripheral sensory neuropathy [68]. Therefore, a more hydrophilic version of ^18^F-FMISO, ^18^F-FETNIM, was developed [68].

**Figure 5 pharmaceuticals-18-00459-f005:**
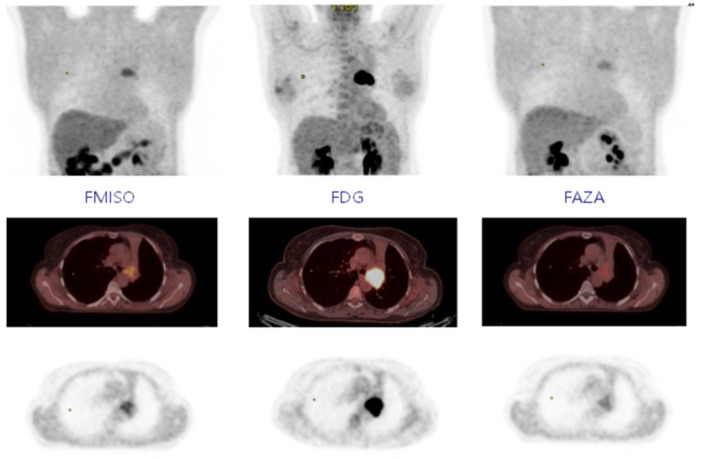
Representative PET imaging results comparing [^18^F]-FMISO, [^18^F]-FDG, and [^18^F]-FAZA in a lung cancer patient. Coronal, axial fused, and axial PET images demonstrate tracer uptake in the tumor, highlighting differences in hypoxia ([^18^F]-FMISO and [^18^F]-FAZA) and glucose metabolism ([^18^F]-FDG). The images show the distinct localization of each tracer, emphasizing the role of hypoxia imaging ([^18^F]-FMISO and [^18^F]-FAZA) versus metabolic imaging ([^18^F]-FDG) in tumor characterization This figure is reproduced from Thureau et al., 2021 under a CC BY 4.0 license (DOI: 10.3390/cancers13164101) [69]. No additional permission is required.

#### 3.3.2. ^18^F-FETNIM

Another type of nitroimidazole called ^18^F, which is labeled fluoroerythronitroimidazole (FETNIM), has been found to be more cost-effective and hydrophilic than FMISO in hypoxic tumor models [70]. Compared with FMISO, ^18^F-FETNIM has shown less penetration of the blood–brain barrier (BBB) while demonstrating similar tumor-to-muscle uptake ratios and correlation with the oxygenation status in mice with C3H mammary carcinomas [71]. Man Hu et al. [72] suggested that ^18^F-FETNIM can be a valuable tool for evaluating tumor hypoxia and targeting tumor cells resistant to conventional therapy in NSCLC patients. However, Yuchun Wei et al. [73] demonstrated that tumors exhibit significantly higher uptake with ^18^F-FMISO compared with ^18^F-FETNIM. Therefore, they suggested that ^18^F-FMISO may be a better hypoxia tracer for lung cancer patients (see Figure 6). On the contrary, some studies comparing the uptake of FMISO and FETNIM in lung cancer patients showed that there is no significant difference between the two in terms of tracer uptake and tumor-to-blood ratio (TBR). Nonetheless, both FETNIM and FMISO tumor uptake are considered effective prognostic factors for a patient’s overall survival [72,73,74].

#### 3.3.3. ^18^F-FDG

Cancer is characterized by the reprogramming of energy metabolism, which has led to the development of tracers like fluorine-18 (^18^F) bound to the glucose analogue 2-[^18^F]fluoro-2-deoxy-D-glucose (^18^F-FDG) being extensively used for diagnostic purposes in various malignant conditions [5,75].

The most common radiotracer used for PET imaging is ^18^F-FDG [75]. This is because cancer cells consume more glucose than normal cells, which makes them easier to identify and locate [76]. A radiolabeled glucose analogue, ^18^F-FDG has the ^18^F radioisotope substitute the hydroxyl group of glucose at position 2. By injecting a small amount of ^18^F-FDG into a patient, the PET scanner can produce images which highlight the areas of the body with the highest glucose uptake [76]. Studies have demonstrated that this radiopharmaceutical is effective in determining the extent of cancer, detecting distant metastases, planning therapy, evaluating responses to treatment, and locating tumors in cases where the primary one is unknown [76,77,78,79]. Substantial evidence suggests that ^18^F-FDG PET/CT should be included in the diagnostic workup of patients with lung cancer due to its ability to provide valuable information for staging the disease and selecting appropriate therapy [80].

Clavo et al. proposed that ^18^F-FDG could be a substitute for hypoxia imaging [80]. This is because glucose metabolism is stimulated via HIF1α under hypoxic conditions. Zhao et al. demonstrated the relationship between the uptake pattern of ^18^F-FDG, the expression of glucose transport proteins and hexokinase, and the presence of HIF-1α using autoradiography and immunostaining [81,82]. Moreover, Dierckx and Van De Wiele discovered a correlation between glucose metabolism and hypoxia, detected using ^18^F-FDG PET imaging [18,83]. Several studies have been conducted to evaluate the relationship between ^18^F-FDG uptake and the level of hypoxia. However, the results obtained from these studies are contradictory to results depicted by Dierckx and Van De Wiele [18,83]. Zimny et al. discovered that ^18^F-FDG PET could not reliably differentiate between hypoxic and normoxic tumors, despite the effect of hypoxia on glucose metabolism in head and neck tumors [84]. Gagel et al. [66] demonstrated that there was no correlation between ^18^F-FDG PET and hypoxia, as assessed using ^18^F-FMISO, in patients with NSCLC. Moreover, the presence of HIF-1α protein in non-hypoxic tumor regions suggests that other factors may indirectly affect glucose metabolism and ^18^F-FDG uptake in these regions [20].

Overall, ^18^F-FDG PET/CT plays a crucial role in evaluating the response to therapy in various tumors, including lung cancer [85]. It has a specificity of 92% and a sensitivity of 100% in NSCLC patients [85]. Some publications suggest that the level of ^18^F-FDG avidity after treatment is linked to worse survival rates and an increased risk of local recurrence as well as distant metastases [86]. Recent studies indicate that interim ^18^F-FDG PET during chemotherapy can predict responses to therapy and patient survival in lung cancer cases [87]. Derlin et al. [88] conducted a study which demonstrated the significant benefits of using PET imaging in patients with(NSCLC. Furthermore, the study found that ^18^F-FDG PET imaging helped avoid unnecessary thoracotomies in about 50% of patients compared with those who did not undergo PET imaging. Therefore, PET/CT should be used in all patients who are planning to undergo radical radiotherapy (RT), surgery, or radio-chemotherapy.

While the ^18^F-FDG radiotracer is commonly used in PET/CT studies, it has a few limitations. Glucose accumulation in inflammatory cells can cause false-positive results in ^18^F-FDG PET/CT scans, especially during lymph node staging, tuberculosis, sarcoidosis, or other pulmonary inflammatory conditions [89]. A study used ^18^F-FDG PET/CT scans to distinguish between tuberculosis and malignant lung lesions, both in early and delayed stages. They reported that dual-phase ^18^F-FDG PET using both early and delayed imaging appears to be useful in distinguishing between TB and malignant lung lesions. In general, endobronchial ultrasound (EBUS) remains the gold standard for distinguishing between malignant and benign lymph nodes [90]. Due to these concerns, it is necessary to investigate more specific PET tracers which could provide additional information in such cases.

#### 3.3.4. ^18^F-FSPG

Over the past decade, researchers have conducted studies to explore alternative changes in energy metabolism as a means of creating tracers which are more specific to cancer cells. The molecule (4S)-4-(3-[^18^F]fluoropropyl)-l-glutamate (^18^F-FSPG), also known as BAY 94-9392, is an ^18^F-labeled glutamic acid derivative designed to target the xc⁻ system, a key player in tumor-specific metabolic adaptations [91].

^18^F-FSPG is transported into cancer cells via the xCT transporter, a cystine/glutamate antiporter which facilitates the exchange of intracellular glutamate for extracellular cystine. This transporter functions as a glutamate/cystine exchanger (SLC7A11/SLC3A2 (CD44) heterodimer) which exchanges L-cystine for L-glutamate across the cell membrane, allowing L-cystine to enter the cell and L-glutamate to exit into the extracellular compartment [92]. In addition, ^18^F-FSPG has shown promising results in preclinical and clinical studies for cancer detection, with potential applications in differentiating benign from malignant lesions [92,93,94,95,96]. Moreover, ^18^F-FSPG has demonstrated utility in detecting cancers such as breast, lung, prostate, colorectal, and head and neck cancers as well as hepatocellular carcinoma, with ongoing research to validate its use in other malignancies [92,93,94,95,96,97,98]. Studies suggest that ^18^F-FSPG may offer advantages over ^18^F-FDG by reducing false positives in benign or inflammatory lesions [92,93,94,95,96,99]. An exploratory study evaluating ^18^F-FSPG PET/CT in different cancers demonstrated a comparable tumor-to-background ratio (TBR) and detection rate between ^18^F-FSPG PET/CT and ^18^F-FDG-PET/CT in NSCLC patients. Significant correlations between ^18^F-FSPG uptake and the xc⁻ system have been demonstrated in certain tumors using immunohistochemical staining [94]. Baek et al. [94] tested the diagnostic potential of ^18^F-FSPG PET/CT in detecting intermediate-risk pulmonary nodules in 26 patients and compared the results with ^18^F-FDG-PET/CT. According to this study’s findings, there was no significant difference in diagnostic accuracy between these two radiopharmaceuticals [94]. Ongoing clinical trials aim to evaluate the diagnostic performance of ^18^F-FSPG, particularly in patients with lung cancer [98,100].

In evaluating indeterminate pulmonary nodules, a study compared the effectiveness of ^18^F-FSPG PET imaging against the standard ^18^F-FDG PET. The findings revealed that ^18^F-FSPG PET successfully identified malignant lesions which exhibited low or no uptake in ^18^F-FDG PET scans. This suggests that ^18^F-FSPG PET can detect certain lung tumors which ^18^F-FDG PET might miss. Therefore, ^18^F-FSPG PET could serve as a complementary imaging modality, enhancing the detection and characterization of lung tumors, especially in cases where ^18^F-FDG PET results are negative or inconclusive (see Figure 7) [91].

#### 3.3.5. ^18^F-FLT

^18^F-FLT is a thymidine analogue used as a radiopharmaceutical for cell proliferation imaging primarily due to its correlation with Ki-67 expression, which is a nuclear protein [101]. While ^18^F-FLT primarily functions as a tracer for cell proliferation by targeting thymidine kinase 1 activity, its use in a hypoxia-related review can be justified by the relationship between hypoxia and tumor proliferation. Tumor hypoxia is a major driver of cancer progression, and hypoxic regions often correlate with increased tumor aggressiveness, resistance to therapy, and enhanced proliferative signaling. These effects make it crucial to consider proliferation imaging alongside hypoxia imaging for a more comprehensive understanding of tumor biology. The concentration of FLT in lung cancer cells reflects the activity of the proliferation-dependent enzyme thymidine kinase, which is upregulated selectively prior to and throughout the S phase (DNA replication and growth phase) [102]. Vera et al. conducted a pilot study with three different tracers, namely ^18^F-FDG for metabolism, ^18^F-FMISO for hypoxia, and ^18^F-FLT proliferation, for five patients with NSCLC who were undergoing radiation therapy [103]. The study utilized the tracers before and during treatment, and the results showed correlations between the three [103].

Alwadani et al. found that ^18^F-FLT had a higher specificity but a low sensitivity compared with ^18^F-FDG for diagnosing and staging patients with lung cancer [104]. They also stated that although this tracer was linked to time to progression and progression-free survival (PFS), it had a weaker correlation with tumor size and overall survival (OS). ^18^F-FLT may not be an ideal ligand for imaging of patients with SCLC because of its limited uptake, as it only reflects cells in the S phase [104]. This could limit its diagnostic performance, since cells in other phases may not be detected. Additionally, the accumulation of FLT in rapidly dividing cancer cells and benign tissues, such as bone marrow and the liver, could also be influenced by cell transport mechanisms [105]. Still, ^18^F-FLT provides predictive values for therapy in other cases. A study showed that radiation therapy significantly reduced ^18^F-FLT uptake in patients with NSCLC, indicating that this radiotracer could potentially detect early radiation response [106]. It was also demonstrated that the uptake of ^18^F-FLT is substantially reduced after receiving 5–11 fractions of RT [106]. Moreover, after completing chemotherapy, Allen et al. observed a considerable reduction in ^18^F-FLT uptake, which had a better diagnostic accuracy than ^18^F-FDG [107]. This also appears to be consistent with the previously mentioned study by Alwadani et al. [104]. FLT is generally not considered an alternative to the current pan-cancer agent FDG. However, it could be used in specific clinical conditions, such as assessing early therapeutic response or monitoring recurrences of NSCLC [104].

#### 3.3.6. ^18^F-FAZA

The use of FMISO is limited due to its slow clearance from normoxic background tissue, leading to low image contrast and prolonged imaging protocols [108,109]. The combination of FMISO’s slow clearance and its radioisotope’s half-life of 110 min poses challenges for efficient imaging [109]. These challenges underscore the need for a more hydrophilic alternative to FMISO, such as ^18^F-FAZA, which addresses these limitations while retaining the ability to detect hypoxic tissue accurately. Subsequently, a second-generation nitroimidazole derivative called ^18^F-FAZA was developed as a follow-up to ^18^F-FMISO [109].

^18^F-FAZA demonstrates enhanced pharmacokinetic properties, including greater hydrophilicity, lower lipophilicity, faster vascular clearance, and renal excretion, making it a more efficient tracer compared with FMISO [108]. These radiotracer characteristics result in higher uptake and improved hypoxia-to-normoxia contrast compared with FMISO, which exhibits slower clearance and is excreted via the hepatobiliary system [108,109]. In a comparison between ^18^F-FAZA and ^18^F-FMISO in two murine tumor models, ^18^F-FAZA consistently showed higher tumor-to-muscle and tumor-to-blood ratios, attributed to its faster clearance rate. Furthermore, ^18^F-FAZA has been clinically applied for imaging hypoxia in a range of tumors, including gliomas, rhabdomyosarcomas, lymphomas, and carcinomas of the lung, head, neck, cervix, and rectum [83,110,111,112,113]. A research study conducted on 38 patients suffering from advanced non-small-cell lung cancer (NSCLC) found that the level of hypoxia, as measured by ^18^F-FAZA, was linked to disease progression during chemoradiation treatment [114]. Emerging evidence suggests that ^18^F-FAZA holds greater potential for clinical and preclinical applications compared with its first-generation counterpart, ^18^F-FMISO [108,115]. Busk et al. [116] suggested that ^18^F-FAZA imaging data may be sufficient for radiotherapy planning. Their study demonstrated the reproducibility of FAZA PET hypoxia imaging at baseline and during fractionated radiotherapy in tumor-bearing mice, highlighting its potential for treatment adaptation. This evidence supports the sufficiency of FAZA imaging data for guiding radiotherapy decisions by assessing tumor hypoxia over time.

Although various clinical studies have been conducted on lung malignancies [111,117,118], none of these clinical studies provide a direct comparison of ^18^F-FAZA with hypoxia biomarkers, to the best of our knowledge. For example, numerous research groups have confirmed the validity of pimonidazole HCl as a hypoxia marker [119,120,121], and indeed, a correlation has been reported between pimonidazole immunostaining in tissues and the level of oxygen as measured with electrodes or luminescent pO_2_ sensors [122]. However, limited studies associate ^18^F-FAZA uptake with hypoxia hallmarks or pimonidazole immunostaining in specific cancer cell xenografts. Some clinical studies have suggested that the uptake of ^18^F-FAZA is dependent on oxygen, but there are only a few studies which validated or established its correlation with oxygenation.

Under ideal circumstances, the uptake of a hypoxia tracer is not directly proportional to perfusion. However, a lack of perfusion can potentially affect tracer delivery, which may disturb the association between tissue hypoxia and ^18^F-FAZA uptake [123]. Subsequently, in low-perfusion regions which contain strongly hypoxic tissue, poor tracer delivery may result in the underestimation of hypoxia as detected by ^18^F-FAZA [123]. The uptake of ^18^F-FAZA is expected to be higher in areas with low perfusion, while regions with high perfusion may exhibit lower uptake due to a sufficient oxygen supply. Unfortunately, most PET hypoxia tracer uptake analysis methods do not account for potential perfusion effects at present [111,124].

#### 3.3.7. ^18^F-EF3

Several other molecules of hypoxia-avid radiopharmaceuticals are represented by ^18^F-2-nitroimidazol-pentafluoropropyl acetamide (^18^F-EF5) and ^18^F-2 nitroimidazol-trifluoropropyl acetamide (^18^F-EF3). The EF family of molecules is promising due to their increased stability in comparison with FMISO [125,126,127,128,129,130,131]. The compound ^18^F-EF3 was thoroughly researched by Mahy et al. [125] for its potential use in ear, nose, and throat (ENT) tumors. Their results revealed that ^18^F-EF3 was eliminated from the body at a faster rate than ^18^F-FMISO. Additionally, the images obtained four hours after injection indicated that the compound was evenly distributed in healthy tissue, being similar to the distribution observed with ^18^F-FMISO. However, the EF family was abandoned due to its complex radio-labeling processes and tendency to rapidly de-fluorinate. Moreover, these compounds have a long biological half-life of 12 h, which can cause them to bind to aerobic cells and remain in the bloodstream, leading to a poor TBR [131]. Many other clinical studies have been carried out using this radiotracer [125,126].

#### 3.3.8. ^18^F-EF5

Another member of the EF family is 2-(2-nitro-1*H*-imidazol-1-yl)-*N*-(2,2,3,3,3-penta-fluoropropyl)-acetamide labeled with fluorine (^18^F-EF5). ^18^F-EF5 has a higher octanol-water partition coefficient, enabling passive diffusion through cell membranes, which could enhance the consistency of both tumor uptake and tracer distribution [126,127]. As seen with ^18^F-EF3, ^18^F-EF5 has demonstrated the capability of identifying tumor hypoxia in head and neck, cervical, and brain cancers [126]. However, again, the complexity of radiolabeling processes gives ^18^F-EF5 no significant advantage over ^18^F-FMISO. The tracer ^18^F-EF5 was used to study NSCLC, and its uptake was found to be correlated with tumor hypoxia [128,129]. Its excretion occurs mainly through the kidneys, with a smaller amount excreted in the liver and bowels [127]. Due to its lipophilicity, ^18^F-EF5 is uniformly distributed in the normal brain [127].

#### 3.3.9. ^18^F-HX4

3-[^18^F]fluoro-2-(4-((2-nitro-1*H*-imidazol-1-yl) methyl)-1*H*-1,2,3-triazol-1-yl)propan-1-ol (^18^F-HX4) is a third-generation 2-nitroimidazole nucleoside analogue [130,131]. Dubois et al. synthesized ^18^F-Flortanidazole (^18^F-HX4) using a process known as “click chemistry”, which involves assembling small units together through heteroatom linkages [131]. This technique enables a faster and more efficient synthesis process. The presence of a 1,2,3-triazole moiety in ^18^F-HX4 increases its hydrophilicity, resulting in improved renal clearance compared with FMISO [130,131]. Despite having favorable pharmacokinetics, the limited published clinical data are a reason for its limited use.

Nonetheless, the initial clinical trials demonstrated similar outcomes when compared with FMISO, provided that image acquisition was performed 4 h after tracer administration [132,133]. Ureba et al. reported that there was a correlation between the level of pO_2_ and ^18^F-HX4 uptake in patients with non-small-cell lung cancer (NSCLC) [134]. Furthermore, another study carried out among patients with NSCLC which explored the relationship between tumor metabolism and tumor oxygenation using ^18^F-FDG and ^18^F-HX4, respectively [135], found that the biological imaging characteristics observed in NSCLC tumors exhibited significant interpatient and intratumor variability. The majority of the tumors showed overlapping regions of hypoxia and metabolic activity, while regions with high blood flow did not align with areas of increased hypoxia. Additionally, Zegers et al. used PET imaging with ^18^F-HX4 in a study of 15 NSCLC patients. They observed that this tracer showed uptake in the majority of NSCLC lesions. Moreover, it has been suggested that ^18^F-HX4 PET could be a more effective hypoxia imaging biomarker than ^18^F-FMISO, as it can be used with a shorter injection-acquisition time [136]. However, further validation in a larger patient cohort is still required. Zegers et al. conducted several studies which revealed the potential application of ^18^F-HX4 as a hypoxia PET tracer in NSCLC and HNSCC. They found that it could be useful for monitoring changes in hypoxic areas during radiotherapy or treatment intensification [137]. One of these studies was conducted on two ^18^F-HX4 PET scans on patients with lung, head, and neck cancer in the same week before radiotherapy treatment and showed that ^18^F-HX4 PET provided consistent and reproducible results over time [138].

This study on [^18^F]HX4 for hypoxia PET imaging in lung cancer also demonstrated that repeated scans provide highly reproducible results, with SUVmax and SUVmean showing strong correlations between scans (r = 0.958 and r = 0.946, respectively). Figure 8 illustrates a voxel-wise analysis of lung cancer patients, highlighting stable hypoxic regions over two scans and enabling reliable mapping for potential hypoxia-targeted therapies. This imaging technique identifies hypoxic tumor areas with spatial consistency, making it a promising tool for personalizing treatments in patients with hypoxic tumors [138].

#### 3.3.10. Cu-ATSM

Thiosemicarbazone-based tracers, such as copper complexes of diacetyl-bis(N4-methylthiosemicarbazone) (Cu-ATSM), are widely used for imaging hypoxia due to their selective accumulation in hypoxic tissues. However, the precise mechanisms underlying the hypoxia selectivity of this tracer group still remain incompletely understood [131]. The most widely accepted theory suggests that Cu-ATSM is initially reduced in both normoxic and hypoxic cells to form a less lipophilic and unstable Cu(I)-ATSM complex. However, in the case of normoxia, this complex undergoes reoxidation, which causes it to be expelled from the cell [130]. Cu-ATSM exhibits favorable pharmacokinetics, with quick uptake in hypoxic tissues and rapid clearance from normoxic tissues [123,139], which enable imaging 30 min after injection.

On the contrary, another study indicated the uncertainty of considering Cu-ATSM as a hypoxia-selective radiopharmaceutical [140]. For instance, the documented relationship between Cu-ATSM and fatty acid synthase (FAS) proves that the former is not capable of detecting hypoxia due to its altered uptake pattern during FAS overexpression associated with prostatic malignancies [141]. Additionally, changes in the intracellular environment and biology (mitochondrial changes and pH, respectively) may lead to non-hypoxic trapping of ^64^Cu-ATSM [140]. A comparative study between ^64^Cu-ATSM and ^18^F-FDG found no correlation between their corresponding uptakes [21]. The accumulation of ^64^Cu-ATSM was observed at the periphery, while a high accumulation of ^18^F-FDG was noticed at the center of the tumor. This difference may be explained by the differences in the perfusion and metabolic demand of the hypo-perfused tumor center (see Figure 9). Furthermore, an inverse correlation between ^64^Cu-ATSM uptake and blood flow has been observed [21].

Cu-ATSM is potentially more useful in clinical practice than FMISO due to its ability to accurately determine the hypoxia conditions within tumor cells [142]. However, several studies indicate that the uptake of Cu-ATSM is influenced not only by hypoxia but also by other factors, such as tumor type and blood flow [123,131,139,143]. This affects the release of copper ions, which in turn affects Cu-ATSM uptake. Clinical trials of Cu-ATSM have shown conflicting results as a hypoxic agent compared with FMISO in lung cancer over the past two decades [130]. Still, Cu-ATSM may have better clinical applicability due to its favorable pharmacokinetics [130]. Cu-ATSM imaging has several important advantages, including its ability to predict treatment response after chemo-radiotherapy, which makes it a useful prognostic factor [22]. Cu-ATSM imaging can, therefore, play a role in treatment planning for advanced stages of lung cancer, as well as for monitoring treatment response [22,108,144]. Studies have shown that Cu-ATSM imaging can be particularly effective in these scenarios.

Copper-labeled radiopharmaceuticals, including ^60^Cu, ^61^Cu, and ^64^Cu, have distinct physical properties which determine their applications. First, ^60^Cu (half-life of ~23.7 min) is suited for imaging rapid physiological processes, such as acute hypoxia, and ^61^Cu (half-life of ~3.33 h) provides a longer imaging window, allowing for the study of processes with intermediate kinetics. Meanwhile, ^64^Cu (half-life of ~12.7 h) is the most versatile of the three, supporting both diagnostic imaging and therapeutic uses due to its longer decay characteristics. Moreover, Cu-PTSM (pyruvaldehyde-bis(*N*^4^-methylthiosemicarbazone)) is specifically designed for perfusion imaging, targeting highly oxygenated and well-perfused tissues. On the contrary, Cu-ATSM compounds preferentially accumulate in the hypoxic regions of tumors, making them valuable for studying oxygen-deprived microenvironments. These radiopharmaceuticals provide complementary insights, enhancing our ability to characterize tumors and tailor treatment strategies [34,145,146,147,148,149,150,151].

Lohith et al. compared the uptake and intra-tumoral distribution of ^62^Cu-ATSM and ^18^F-FDG in squamous cell carcinoma (SCC) and adenocarcinoma, and the findings revealed that SCC showed spatial mismatching of the two tracers, with ^62^Cu-ATSM accumulating in peripheral regions and ^18^F-FDG concentrating in central regions, whereas in adenocarcinomas, there was a more homogeneous distribution. These variations highlight the potential of dual-tracer PET imaging for identifying tumor hypoxia and glucose metabolism differences, offering valuable insights for diagnosis and treatment planning (see Figure 8) [152].

**Figure 9 pharmaceuticals-18-00459-f009:**
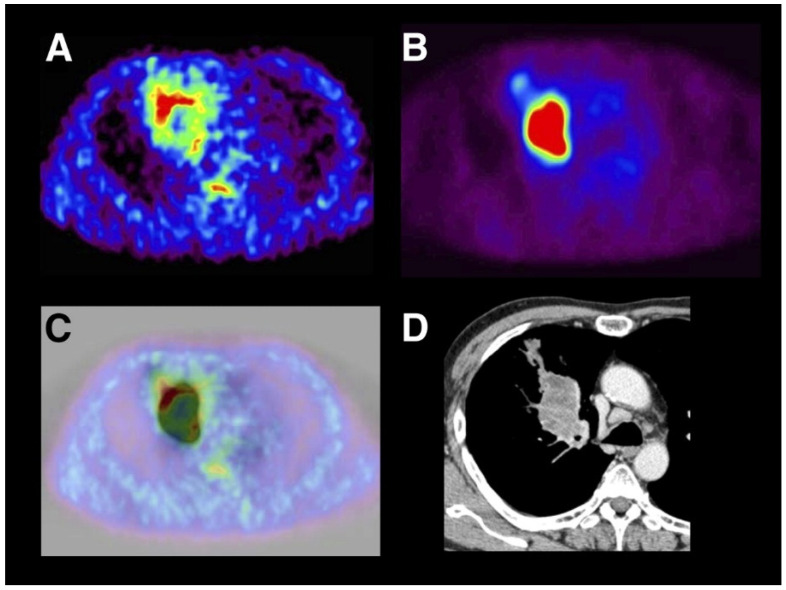
Transaxial PET images of ^62^Cu-ATSM (**A**), ^18^F-FDG (**B**), their fused overlay (**C**), and the corresponding CT scan (**D**) of a 59-year-old male with SCC in the right hilus. The fusion image displays ^62^Cu-ATSM PET in color and ^18^F-FDG PET in grayscale. Reproduced from Lohith et al., 2009 under a CC BY license (DOI: 10.2967/jnumed.109.069021) [152].

## 4. Discussion

It is important for a design of hypoxia agents to follow the general medicinal practice for radiopharmaceuticals and also exhibit various favorable characteristics. This article explores the potential of certain PET radiotracers and their corresponding hypoxia imaging. However, it is imperative to understand the major limitations of traditional methods to measure hypoxia, such as immunohistochemical (IHC) assays. IHC assays are considered the gold standard for hypoxia detection, and nitrimidazoles are the extrinsic markers used for this purpose [153,154,155]. During hypoxia, the nitrimidazoles become trapped inside cells due to the predominance of covenant bonding with the cells’ macromolecules. A subsequent biopsy should be taken, and employing antibody markers to the biopsy specimen is required to acquire the fluorescent signal. This process of using IHC assays is difficult, cumbersome, invasive, and operator-dependent. Other examples of measuring hypoxia include protein-based assays which use endogenous marker proteins, such as carbonic anhydrase IX and glucose transporter 1, to identify the oxygen levels by detecting overexpression of hypoxia-related proteins [156]. This assay again has similar limitations to those mentioned for IHC [157,158].

Targeting specific molecular processes associated with hypoxia, rather than assessing overall organ activity, provides a more specific and insightful understanding of tumor development. This targeted approach enables earlier detection and more precise characterization of the tumor microenvironment, ultimately improving diagnostic accuracy and treatment strategies [159,160,161]. However, it is important to carefully consider various factors when selecting the radionuclide, biomolecule, and chelation agents [162]. These factors have a direct impact on the specific binding, in vivo affinity, metabolic stability, overall pharmacokinetics, and bio-behavior of the product molecule. Therefore, the design of a hypoxia agent must undergo an optimization process to ensure the best imaging criteria, such as high specific binding and selectivity [163]. This involves choosing the most suitable radioisotope, labeling site, lipophilicity, and size of the targeting molecule. When selecting a radioisotope, one must consider the energy, half-life, and traveling range of the β+ particles emitted from the isotope [163]. Zirconium-89 (Zr) is used in hypoxia imaging due to its long half-life, compatibility with PET imaging, and ability to label hypoxia-targeted biomolecules such as CAIX inhibitors and antibodies, enabling precise visualization of hypoxic tumor microenvironments [164]. However, more energetic isotopes (such as 89Zr with a half-life of 78.4 h, E_mean_ of 0.396 Mev, and mean range of 1.3 mm) travel further in the tissue, which can affect the spatial resolution. After designing the radio-molecule, it should undergo thorough studies to test its feasibility and applicability (see Figure 10).

### 4.1. Technical Consideration

Effective hypoxia imaging relies on two key qualifications: imaging modality and hypoxia delineation. The chosen modality must possess the sensitivity to detect subtle variations in tumor and tissue oxygenation, ideally enabling (1) differentiation of hypoxia from other oxygen-related pathologies; (2) the estimation and identification of cellular rather than vascular pO_2_ values; and (3) sensitivity within the low mmHg range (0–15 mmHg). In the context of hypoxia imaging, it is essential for the imaging equipment to meet specific technical specifications, including an appropriate spatial resolution (70–100 µm) and temporal resolution [165,166,167]. These resolutions should be equal to or smaller than the distribution and diffusion distances of the imaging agents [168,169]. Furthermore, the imaging modality must facilitate repeated studies across various regions of the body with minimal invasiveness. Positron emission tomography (PET) generally fulfills these criteria and enables real-time imaging through the use of various radiotracers. PET operates by detecting gamma ray signals emitted from the radiotracer retained in the tissue, which correlates with tissue oxygenation. The temporal heterogeneity in a tumor’s blood supply is a limiting factor in hypoxia imaging [170,171,172], as well as dynamic tumor vasculature [170,171,172], presenting another challenge during PET imaging.

Tumor-associated blood vessels often exhibit deformities and abnormalities [172], resulting in longitudinal and radial variations in the blood supply which can change within seconds [173]. This necessitates a higher temporal resolution and direct measurements to accurately assess these fluctuations. Moreover, the continuous remodeling of a tumor’s vasculature leads to variations in red blood cell flux within the micro-vessels lasting from minutes to hours, which can significantly impact oxygen saturation [174,175].

In order to overcome these limitations, PET can be integrated with other imaging modalities to provide a more comprehensive analysis. Magnetic resonance imaging (MRI), utilizing distinct principles for imaging hypoxia, is a promising candidate for addressing these challenges [139]. MRI can offer a higher temporal resolution and direct measurements of the partial oxygen pressure (pO_2_), such as through fluorine-19 magnetic resonance spectroscopy (MRS). This technique allows for the assessment of direct oxygen consumption by detecting changes in the probe (e.g., perfluorocarbons) during its interaction with O_2_. MRS can quantify pO_2_ levels in a range higher than PET (0–100 mmHg) with high sensitivity (1–3 mmHg) [176,177,178]. Although the cost of MRS is a notable limitation, it can be complemented by various other imaging techniques, including immunohistochemistry, reporter gene imaging, oxygen-sensing probes, optical spectrometry, and electron paramagnetic resonance imaging [179].

Quantification of hypoxia agent levels is typically achieved using one of three parameters: the standard uptake value (SUV), tumor-to-blood ratio (TBR), or hypoxia fraction (HF). The SUV is a numerical representation of tumor uptake commonly measured with ^18^F-FDG following the administration of a radioactive dose [19,48]. This value is normalized to the injected dose and distribution factors, such as body weight, and is calculated using the following formula:$$SUV=\frac{ACvoi\;(\frac{kBq}{mL})}{radiactive\;dose\;(\frac{MBq}{BW\;\left(kg\right)})}$$
where, ACvoi represents the concentration of the average activity and the administered FDG dose’s (decay-corrected) activity. Comparing the four main hypoxia radiotracers (i.e., ^18^F-FAZA, ^18^F-HX4, ^18^F-FMISO, and ^64^Cu-ATSM). it can be found that each radiotracer shows variation in its SUVs. Therefore, the use of other parameters is necessary to accurately measure hypoxia.

The target-to-background ratio (TBR) refers to the ratio of the maximum radiotracer concentration in the region of interest (ROI) to the background concentration [180]. A tumor-to-blood ratio (TBR) value greater than 1.2 (or 1.4 in some studies) two hours post injection has been established as a hypoxia-specific value for ^18^F-FMISO [19]. However, normal retention of the radiotracer does not allow for imaging at this time, necessitating a waiting period for better TBR estimation. On the contrary, the hypoxia fraction (HF) is defined as the fraction of pixels with a TBR greater than 1.2 (or 1.4) within the region of interest (ROI) at two hours post-injection [19]. Factors affecting standard uptake value (SUV), TBR, and HF quantification can be divided into two main categories: (1) physical and technical factors and (2) physiological effects.

In the former category, several factors recognized in the literature are common sources of errors during SUV, TBR, and HF estimation. One major source of error is the cross-calibration between PET-CT (or PET) and the dose calibrator. Calibration establishes the relationship between the measured count rate from the body and the actual concentration of activity. Failure to fulfill this requirement results in cross-calibration errors, which can introduce a systematic error in the SUV, TBR, and HF equivalent to the relative value of miscalibration [181,182]. To avoid this error, basic calibration should be performed using a ^68^Ga phantom of homogeneous activity in a solid matrix for each mode of data acquisition. The activity used in the calibration (in the phantom and matrix) should be verified using the on-site dose calibrator. This is particularly important in multi-center studies aiming to acquire quantitative data, with the accuracy required to be at least 5–10% [183,184,185].

To ensure accurate dosages for patients or test subjects, estimating the exact dose administered is crucial. This can be achieved by counting the residual activity in the syringe. Synchronizing the PET and dose calibrator clocks accurately is also essential; incorrect synchronization can lead to erroneous estimation of the SUV, TBR, and HF. For decay correction, the calibration time of the dose should be used instead of the injection time. Using the injection time can overlook the interval between calibration and injection, leading to inaccurate calculations of actual activity and the SUV, TBR, and HF. This discrepancy can significantly impact the results, especially with short-lived radionuclides like ^18^F (T_1/2_ = 110 min) [185,186]. Additionally, to prevent infiltration of the dose, a secure cannula should be in place prior to injection, and any residual activity at the injection site should be avoided under the camera, as this may affect the SUV by up to 50% [108].

In the latter category, physiological factors can also influence SUV, TBR, and HF estimation, some of which may originate from the patient or subject. For instance, increased blood glucose levels have been shown to reduce the SUV during ^18^F-FDG imaging. Busing et al. found that changes in blood glucose levels, insulin resistance, diabetes, and obesity could affect SUV estimation. Moreover, the time interval between injection and image acquisition may also impact SUVs and the HF [181]. Lowe et al. discovered that the optimal time for improved SUV estimation and better differentiation between normal and abnormal tissue in pulmonary malignancy is 50 min post injection. However, this may not hold true for all cancer types or other radiopharmaceuticals [141]. Some publications suggest a waiting period of 2–4 h for nitroimidazole analogues and less than 1 h for Cu-ATSM. Exceeding the recommended duration may lead to incorrect SUVs, TBRs, and HFs. Other factors, such as patient breathing and inflammation, may also falsely increase the SUV [141].

It is worth noting that several studies have emphasized the importance of standardizing the appropriate waiting period after chemotherapy and radiotherapy. For chemotherapy, Juweid et al. suggested a minimum waiting time of 14 days [187]. For radiotherapy, others recommended waiting up to three months due to the extent of inflammation following treatment and the involvement of different tissues during therapy. Body weight and height should be considered during every scan to enhance normalization and improve SUV and HF estimation [186,188].

Technical factors play an essential role in accurately measuring hypoxia. The body’s increased noise level can lead to overestimation of SUVs and poor image quality. To counter this problem, it is important to follow good injection (and dose) techniques, appropriate uptake periods, and adequate scan durations [189,190]. Breathing [191,192,193], contrast agents [194,195,196,197], and movement [193,194] during a scan can cause problems in PET/CT image registration, which can affect SUV, TBR, and HR estimation [186,189]. Therefore, proper breath-holding protocols [191] and reconstruction methods (see Figure 11), including attenuation and decay correction, should be used [186]. Increased iterations may also result in increased noise and SUVs. Various reconstruction recommendations have been extensively discussed in the literature [191,192,193,194]. It is advisable to match the image resolutions in multi-center studies, as different image resolutions make the comparison (pre- and post-therapy) of SUVs difficult. Furthermore, standardizing the region of interest (ROI) techniques using a universal or highly automated ROI delineation method may also help reduce sources of uncertainty. Table 2 summarizes the recommendations to enhance standardized uptake values (SUVs) in hypoxia PET imaging.

To measure the tumor volume, there are two methods which can be employed. The first method involves predefining the primary and secondary gross tumor volume (GTV1 and GTV2, respectively) and determining the uptake in the GTV1. The second method involves calculating the DICE coefficient by defining the high percentile uptake value UV (e.g., 60–90% range) at two and four hours post-injection. Another important step is to set the HF threshold (e.g., 6%) and define the volumes at four hours post-injection before comparing them to the values at two hours post-injection. The equation can be described as follows:(1)$$DICE=\frac{UV\;\left(2\;h\right)⋂UV\;(4\;h)}{UV\;\left(2\;h\right)+UV\;(4\;h)}$$

The use of standardized protocols in PET comparison studies is mandatory for achieving the inter-interpretability of hypoxia trials. However, this is still a topic of debate and cannot be standardized easily due to multiple reasons. Firstly, the quality of images is not solely dependent on the patient or subject’s weight and may require the consideration of other factors such as the scanning mode and time per bed position. Secondly, the acquisition and reconstruction parameters are usually preset and fixed by the manufacturer, and adjusting them requires careful consideration [199]. For instance, when changing the parameters for PET acquisition, the CT parameters should be adjusted accordingly. Furthermore, available software for SUV quantification from different companies should be also taken into consideration [186].

The complexity of the reconstruction algorithm presents a challenge for imaging technologists who require educational and practical sessions to bridge this gap. A challenge observed in PET centers in nuclear medicine departments around the world is the lack of proper coordination between the radio-pharmacy and cyclotron units. The initial activity generated from the cyclotron is transferred to the radio-pharmacy unit for counting and recording and subsequently to the PET imaging unit for injections. This may cause uncertainties in the actual activity calculations. This is considered a major problem when the used dose calibrators are not accurately calibrated. Another limitation is the CT setting, which is typically preset and fixed. This may differ from center to center and cause issues with different attenuation correction and standardized uptake values. Furthermore, the lack of proper standardized protocols for PET/CT studies is still an issue. Some centers use diluted oral contrast agents in PET/CT studies, and the way they are used is arbitrary. Many studies showed that the contrast agent may affect PET/CT quantification studies by factors of up to 103%. Therefore, a complete history of the patient’s former medical and radiological procedures should be available. Another subjective issue is selection of the hypoxia cutoff, which seems to differ between many PET studies due to the use of different comparison modalities. This adds more complexity and consequently results in complications in retrospective studies.

A possible solution for addressing this problem is to establish a standard protocol which outlines the technical approach and specifies the gold standard for PET comparisons. Hiring qualified technologists can help minimize the risk of errors during PET imaging. It is important to note that studies investigating the same hypothesis should ideally use the same PET imager and dose calibrator. However, if different PET imagers are used, then the studies should be reregistered and normalized before any comparative analysis takes place.

### 4.2. Tumor Perfusion Versus Uptake

Does a higher level of blood perfusion result in enhanced delivery of drugs or radiotracers? Well, the answer is not simple and depends on the specific radiotracer being used and the surrounding tissues, including the endothelial. Tumor blood flow and hypoxia significantly influence the calculation of standardized uptake values (SUVs) in positron emission tomography (PET) imaging, particularly when assessing tumor metabolism and response to therapies. The chaotic microcirculation within tumors, characterized by irregular blood vessel formation and increased capillary transit time heterogeneity (CTH) [200,201], complicates the relationship between blood flow and the extraction of oxygen and glucose. Elevated CTH can lead to ineffective oxygen extraction, as some blood flows too quickly through capillaries, preventing adequate diffusion of oxygen and nutrients into tumor cells [202]. This results in a state of hypoxia, which is often associated with aggressive tumor behavior and resistance to treatment [11]. Interestingly, while antiangiogenic therapies aim to normalize tumor vasculature and improve oxygenation, studies have shown mixed results; some reported improved tumor oxygenation [201,203], whereas others indicated that these therapies can exacerbate hypoxia. The variability in tumor blood flow (TBF) [19] and oxygen extraction fraction [204] further complicates SUV calculations, as the SUV is influenced by the availability of the radiotracer, which is affected by both blood flow and the tumor’s metabolic state. Therefore, understanding the interplay between tumor blood flow, hypoxia, and SUVs is crucial for accurately interpreting PET imaging results and optimizing therapeutic strategies.

Many theoretical models of hypoxia did not show the full relationship between tumor vasculature and tissue oxygenation [35,200,201,202,203]. It is imperative to understand this complex relationship, as it creates a need to pre-delineate the area where hypoxia is predicted to avoid any overlap between malignant and non-malignant hypoxia. One way to achieve this is by using hybrid modalities for the delineation and localization of tumors to correlate them with hypoxia. However, these innovative methods may face challenges when it comes to the tumor microenvironment.

Many publications suggest that the tumor microenvironment plays a significant role in drug and radiotracer delivery, as well as drug resistance [10,11,205]. The microenvironment of a solid tumor is highly diverse, consisting of varied numbers of cells and phenotypes, and it has an organ-like structure which differs according to the genotype, location, and stage of the tumor. These factors, among others, can impact the delivery of drugs or radiotracers, causing heterogeneity in tumor uptake [26,170]. The sensitivity of constituent cells to different drugs may vary based on the three-dimensional structure of the solid tumor. This may be due to the physics of blood–tissue interactions. For instance, fluid kinematics cause the blood passing through capillaries to experience different types of forces which can alter the microenvironment. Mathematical models can help us study this relationship extensively to understand the bio-behavior of hypoxia radiotracers. Therefore, considering blood dynamics while designing mathematical models may also be beneficial [26,170,206,207].

## 5. Limitations and Future Perspective

Despite the growing utility of PET imaging in hypoxia detection, several limitations must be acknowledged. One major challenge is the variability in tracer uptake, which can be influenced by factors such as blood flow heterogeneity, tumor microenvironment dynamics, and radiotracer pharmacokinetics. The spatial resolution of PET, though adequate for many clinical applications, is still limited when assessing hypoxia at a microscopic level, particularly in tumors with highly heterogeneous oxygenation. Additionally, the reliance on indirect hypoxia biomarkers, such as nitroimidazole-based tracers, introduces potential confounders, as their accumulation can be affected by factors beyond hypoxia, including redox status and cellular metabolism. Standardization of imaging protocols remains a critical issue, as differences in acquisition parameters, quantification methods (SUV, TBR, and HF), and patient-specific physiological variations can impact reproducibility and inter-study comparisons.

To address these challenges, future advancements in PET imaging should focus on improving the spatial resolution and sensitivity through novel detector technologies and improved reconstruction algorithms. Hybrid imaging modalities, such as PET/MRI, offer a promising avenue by combining PET’s functional imaging capabilities with MRI’s superior soft-tissue contrast and oxygenation quantification. The development of next-generation hypoxia tracers with enhanced specificity, faster clearance, and improved target-to-background ratios is also essential. Additionally, artificial intelligence (AI) and machine learning algorithms could play a pivotal role in optimizing image analysis, automating tumor segmentation, and enhancing quantitative hypoxia assessment. The integration of PET with molecular and genetic profiling may also enable a more comprehensive understanding of tumor hypoxia, paving the way for personalized treatment strategies. Ultimately, continued research and technical advancements will be key to refining PET-based hypoxia imaging and expanding its clinical applications in oncology and beyond.

## 6. Conclusions

The ability to accurately detect and quantify tumor hypoxia is critical for optimizing cancer treatment strategies. Improved hypoxia imaging can play a pivotal role in the selection of chemotherapeutic agents which effectively target hypoxic tumor cells, as well as enhancing the effectiveness of radiation therapy, which relies on oxygen availability for maximum therapeutic impact. A key consideration in hypoxia imaging is the distinction between acute and chronic hypoxia. While acute hypoxia can arise from transient changes in blood flow, chronic hypoxia is more relevant to tumor progression and treatment resistance. The focus of imaging should therefore be directed toward chronic hypoxia, which is associated with sustained metabolic and genetic adaptations that contribute to therapeutic resistance. Biomarkers associated with hypoxia, such as HIF-1α, provide valuable insight into these adaptations and should be considered in hypoxia-targeted imaging and treatment planning.

To further enhance the accuracy and clinical applicability of hypoxia PET imaging, standardization of imaging protocols and mathematical modeling are essential. Current challenges include variability in tracer kinetics, differences in PET acquisition and analysis methods, and the need for robust hypoxia thresholds. Addressing these challenges requires international standardization efforts (e.g., the image biomarker standardization initiative), and this should be thoroughly discussed in conferences, training workshops, and continuous updates in nuclear medicine research. Collaborative efforts between oncologists, radiopharmacists, and imaging specialists will be necessary to refine PET imaging techniques and expand their role in personalized cancer therapy.

Moving forward, the integration of advanced imaging modalities, improved radiotracers, and AI-based analytical tools will be key to overcoming the current limitations in hypoxia imaging. These advancements will not only enhance the diagnostic accuracy but also support precision medicine approaches, ultimately leading to better patient outcomes in lung cancer and other malignancies.

## Figures and Tables

**Figure 1 pharmaceuticals-18-00459-f001:**
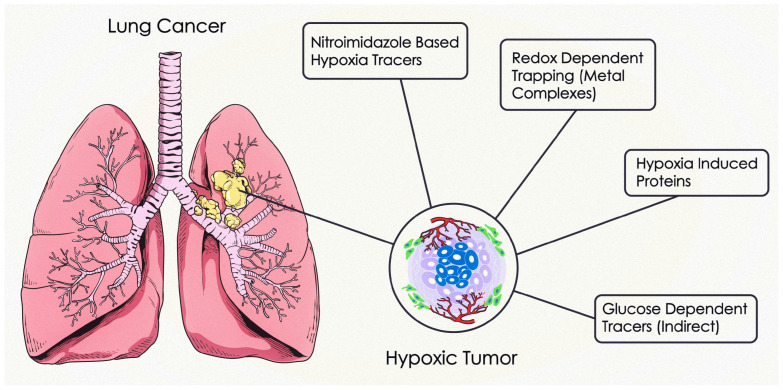
Lung cancer can be detected via biomolecular processes used in lung cancer hypoxia imaging.

**Figure 2 pharmaceuticals-18-00459-f002:**
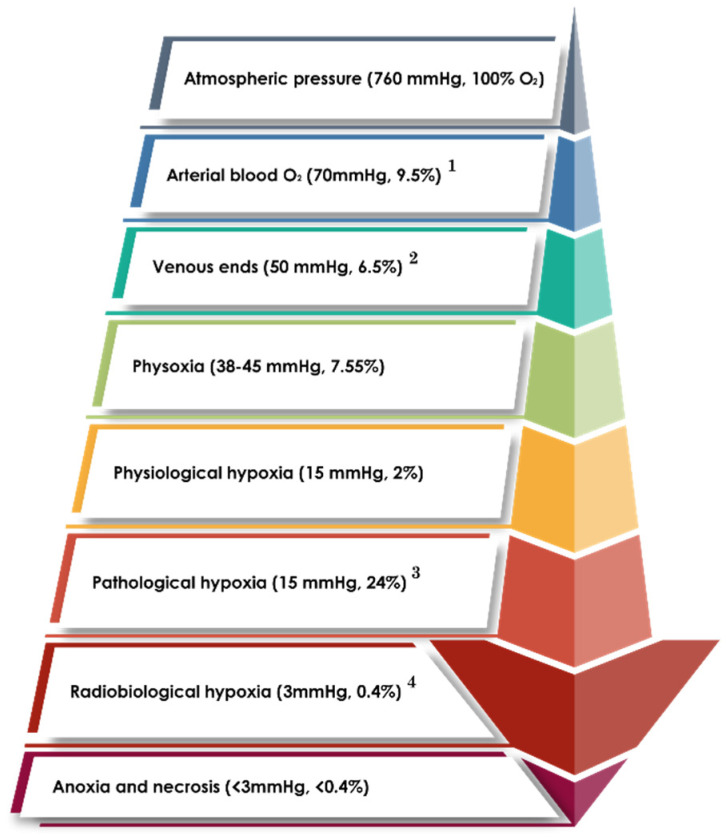
The mean (approximate) oxygen levels in various tissues (pressure in mmHg and oxygen as a percentage). The information included in this figure was derived from several articles [35,36,37]. ^1^: Includes human and animal studies. ^2^: Pressure and percentage of oxygen may vary as they are subject to change due their proximity to major organs. ^3^: Represent disruption to normal homeostasis at or below which a hypoxic catastrophic cascade of the event may initiate (as shown in Figure 3). ^4^: At this point, the radiotoxic effect is at half maximal. Also, the hypoxia cutoff is different for different radiotracers.

**Figure 3 pharmaceuticals-18-00459-f003:**
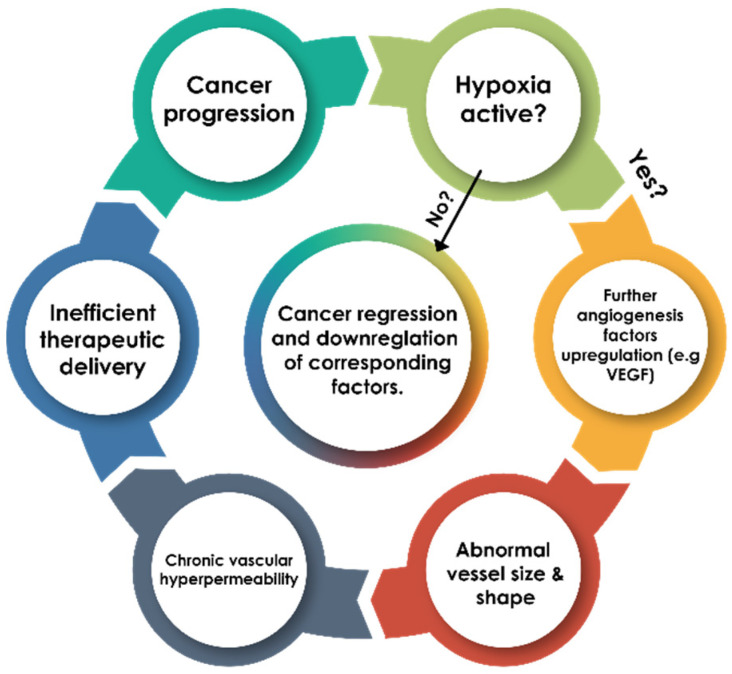
Illustration of the cascade of events leading to cancer progression during therapeutic interventions. The author assumes hypoxia as a starting point of this catastrophic cascade of illness.

**Figure 4 pharmaceuticals-18-00459-f004:**
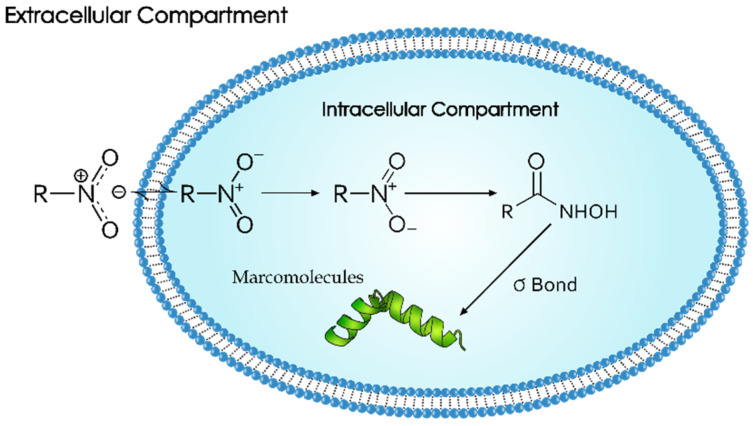
Nitroimidazoles reaction inside hypoxic cells (mechanism of trapping). During the hypoxic conditions, the reactive species (R-NHOH) predominantly form covenant (σ) bonds with intracellular macromolecules and are trapped inside the hypoxic cell.

**Figure 6 pharmaceuticals-18-00459-f006:**
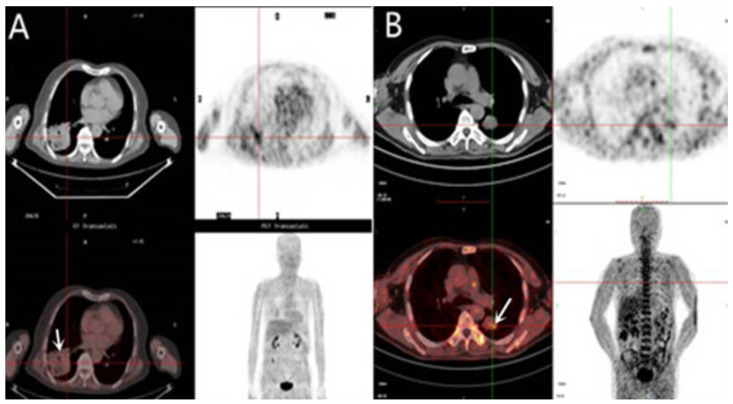
Uptake of ^18^F-FETNIM (**A**) in a squamous carcinoma patient and ^18^F-FMISO (**B**) in a small cell lung cancer patient at 2 h post-injection. The arrows indicate the tumor’s location. Reproduced from Wei et al., 2016 under a CC BY license (DOI: 10.1371/journal.pone.0157606) [73].

**Figure 7 pharmaceuticals-18-00459-f007:**
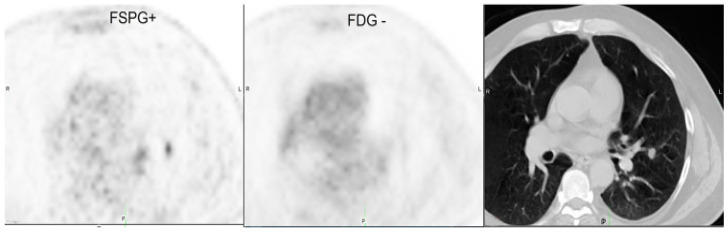
Subject demonstrated positive ^18^F-FSPG uptake (FSBG+) (SUVmax 2.02) and negative ^18^F-FDG uptake (FDG−) (SUVmax 0.7) in PET/CT imaging. Reproduced from Paez et al., 2022 under a CC BY license (DOI: 10.1371/journal.pone.0265427) [91].

**Figure 8 pharmaceuticals-18-00459-f008:**
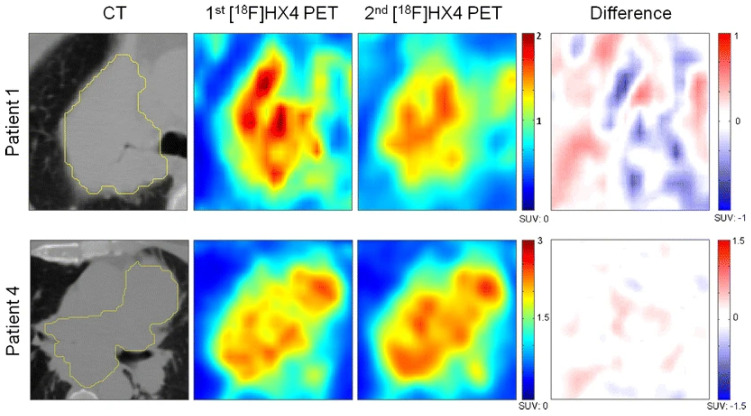
Illustration of voxel-wise analysis in lung cancer patients (patients 1 and 4), showing the axial plane of CT scans with gross tumor volumes outlined in yellow alongside the first and second registered [^18^F]HX4 PET scans and a difference map highlighting variations between the two scans. Reproduced from Zegers et al., 2015 under a CC BY license (DOI: 10.1007/s00259-015-3100-z) [138].

**Figure 10 pharmaceuticals-18-00459-f010:**
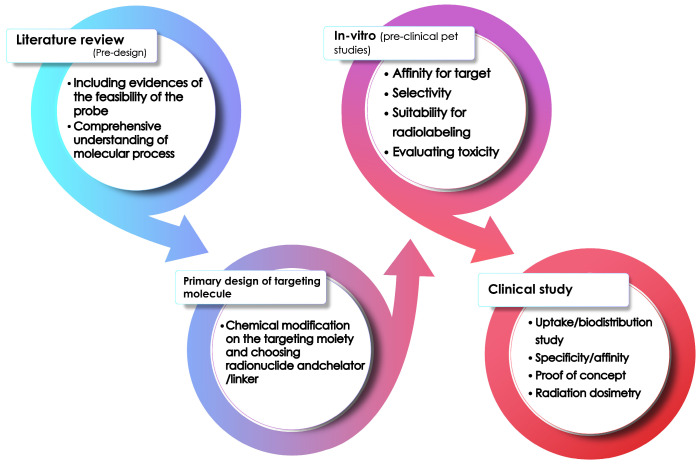
Steps for radiotracer feasibility studies.

**Figure 11 pharmaceuticals-18-00459-f011:**
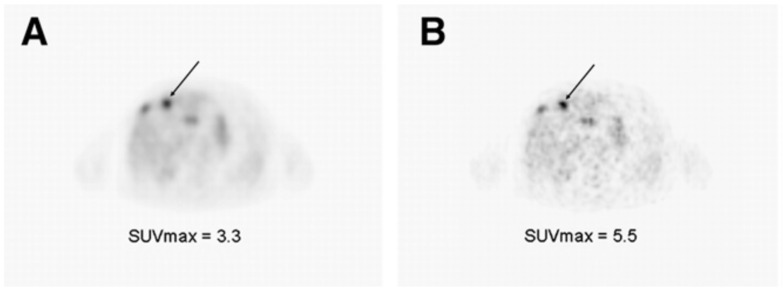
Effect of reconstruction on SUVmax and lesion volume (liver lesion, arrow). (**A**) 2 × 8 OSEM, 128 × 128 matrix, 8-mm filter: ~11-mm resolution, 4.5 mL lesion. (**B**) 4 × 16 OSEM, 256 × 256 matrix, 5-mm filter: ~7-mm resolution, 1.5 mL lesion. (Image from [198]).

**Table 1 pharmaceuticals-18-00459-t001:** This table provides a comprehensive overview of the radiopharmaceuticals specifically utilized for tumor PET hypoxia imaging. It details each compound’s chemical structure, mechanism of uptake, biochemical characteristics (lipophilicity vs. hydrophilicity), and pharmacokinetic and pharmacodynamic profiles.

Radiopharmaceutical	Mechanism of Uptake	Biochemical Characteristics	3D Conformer	Pharmacokinetics and Pharmacodynamics
^18^F-Fluoromisonidazole (^18^F-FMISO)	Under hypoxic conditions, FMISO undergoes bioreduction, forming reactive intermediates which bind to intracellular macromolecules, trapping the tracer in hypoxic cells.	Moderately lipophilic	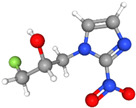	Slow clearance from normoxic tissues, leading to modest tumor-to-background contrast. Primarily excreted via the kidneys and liver.
^18^F-Fluoroazomycin Arabinoside (^18^F-FAZA)	Reduced in hypoxic cells, leading to covalent binding and retention in hypoxic tissues.	More hydrophilic than FMISO	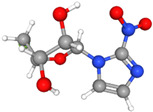	Faster clearance from non-hypoxic tissues than FMISO, improving imaging contrast. Excreted primarily through the renal pathway.
^18^F-EF3	Hypoxia-specific reduction and trapping, resulting in accumulation in hypoxic tumor cells.	Moderately lipophilic	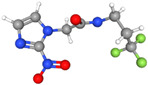	Retained in hypoxic tissues with effective clearance from normoxic areas. Excreted primarily via the renal system.
^18^F-EF5	Bioreduced in hypoxic cells, forming stable adducts retained within hypoxic tumor regions.	Moderately lipophilic	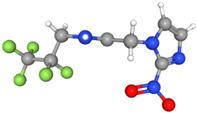	High retention in hypoxic tissues but limited by slow plasma clearance as a result of its lipophilicity. Excreted via the renal system.
^18^F-FSPG (4-(3-[^18^F]Fluoropropyl)-l-glutamate)	Targets the xC^−^ cystine/glutamate antiporter. Upregulated in hypoxic and metabolically active tumor cells.	Hydrophilic	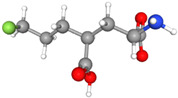	Rapid blood clearance and renal excretion. Provides specific imaging of metabolic and hypoxic regions.
^64^Cu-ATSM	Reduced and trapped within hypoxic cells, selectively accumulating in hypoxic tumor areas.	Lipophilic	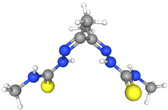	Rapid clearance from blood and high tumor uptake in hypoxic regions. Primarily metabolized in the liver and excreted via the biliary route.
^18^F-FETNIM (Fluoroerythronitroimidazole)	Undergoes bioreduction in hypoxic cells, forming stable adducts retained in hypoxic tissues.	Hydrophilic	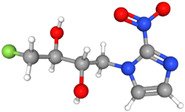	Rapid clearance from normoxic tissues and prolonged retention in hypoxic regions. Primarily excreted via the renal system.
[^18^F]HX4	Similar to FMISO, HX4 is bioreduced in hypoxic cells, resulting in covalent binding and retention in hypoxic regions.	More hydrophilic than FMISO	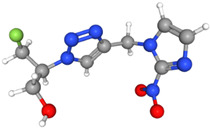	Rapid clearance from normoxic tissues. High retention in hypoxic areas, providing excellent tumor-to-background contrast. Excreted via kidneys.

**Table 2 pharmaceuticals-18-00459-t002:** Recommendations to enhance standardized uptake value (SUV) in hypoxia PET imaging.

Category	Key Points
Recommendations	Use the same scanner and protocols for baseline and follow-up hypoxia PET/CT studies. Ensure precise calibration of dose calibrators for low-activity tracers. Adopt standardized postinjection uptake times specific to the hypoxia tracer used. Correct for respiratory motion in thoracic imaging to improve SUV accuracy. Document tumor oxygenation context and perfusion characteristics to aid interpretation. Avoid confounding factors such as recent oxygen supplementation or hyperbaric oxygen therapy. Provide explicit reporting of region-of-interest (ROI) definitions and imaging protocols for reproducibility.

## Data Availability

Not applicable.

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
