# Peer review of "Hypoxia Imaging in Lung Cancer: A PET-Based Narrative Review for Clinicians and Researchers"

_pharmaceuticals, 2025, doi:10.3390/ph18040459_

Round 1

Reviewer 1 Report

Comments and Suggestions for Authors

The submitted review article present an interesting insight into the the possibility to use hypoxia imaging for lung malignancy. Although the review is quite well written and clear in its explanations, authors should consider some modifications to improve the quality. I am confident that the article can deserve publication in this journal. Below my comments:

-introduction should be more incisive by adding a picture resuming the discussion

-materials authors claim that they conducted a systematic search, but this is not a systematic review. Considering that the inclusion criteria are not properly formulated and the workflow of the search is not clear. Moreover, no tools for systematic search have been used and authors processed articles based probably on their expertise and interest. Obviously this approach is arbitrary and should be affected by some bias based on the users reading articles that consider articles relevant. More explanation in this sense is appreciated, and I suggest to delete systematic and specify the type of review in absence of a real systematic search (i.e. narrative review? or similar).

-Authors should better describe the potential limitation of the study but also of the method (PET imaging for hypoxia) and possible future perspective with the use of more sophisticated approaches or by improving PET imaging. A paragraph in this sense before the conclusion is warmly requested.

-Figure 9 should be enlarged to improve readability.

Comments on the Quality of English Language

moderate change

Author Response

Response to Reviewer 1

We sincerely appreciate your time and thoughtful feedback. Your insights have greatly helped refine our manuscript. We have carefully considered each suggestion and made revisions to improve clarity, accuracy, and presentation. Below, we provide detailed responses and outline the modifications made. Please note that cited line numbers refer to the track changes version (author-coverletter-44849606.v1.pdf).

Reviewer comment 1:introduction should be more incisive by adding a picture resuming the discussion.

Response 1: Thank you for your suggestion. I have now added a figure that classifies the biomolecular processes used in lung cancer hypoxia detection to enhance clarity (line 165-168) . Additionally, I will revise the introduction to be more incisive by incorporating a summarizing image that visually encapsulates the discussion. Please let me know if you have any further recommendations. We added this line in the end of the introduction:

“Figure 1 classifies the biomolecular processes used in lung cancer hypoxia detection.”

Reviewer comment 2: materials authors claim that they conducted a systematic search, but this is not a systematic review. Considering that the inclusion criteria are not properly formulated and the workflow of the search is not clear. Moreover, no tools for systematic search have been used and authors processed articles based probably on their expertise and interest. Obviously this approach is arbitrary and should be affected by some bias based on the users reading articles that consider articles relevant. More explanation in this sense is appreciated, and I suggest to delete systematic and specify the type of review in absence of a real systematic search (i.e. narrative review? or similar).

Response 2: Thank you for your detailed feedback. I acknowledge that the initial description of the methodology suggested a systematic review, while the approach taken aligns more with a narrative review. To address this, I have revised the manuscript to clearly specify that it is a narrative review rather than a systematic one. I have also adjusted the methodology section to provide a more transparent explanation of the article selection process, ensuring clarity about the criteria and approach used. Additionally, I have removed references to systematic tools and workflow to avoid any misrepresentation. Please let me know if further clarification is needed. We added this text to not confuse the reader (Line 193-224):

A broad literature search was conducted to explore the use of PET for assessing hypoxia in cancer. Relevant studies published between 1990 and 2025 were considered, with a focus on original research and comprehensive reviews discussing imaging modalities for hypoxia detection. Studies that did not specifically address hypoxia imaging in cancer, lacked methodological detail, or were limited to commentaries, editorials, or conference abstracts without full-text availability were not included in the discussion.

The literature search identified numerous studies (approximately 730 articles) related to PET, PET/CT, and PET/MR imaging. Articles were selected based on their relevance to the topic, emphasizing key findings and advancements in hypoxia imaging. The selection process was guided by expertise in nuclear medicine, radiopharmacy, and imaging sciences, ensuring a well-rounded perspective. Key studies that contributed significant insights into hypoxia detection were highlighted and discussed in the review.”

Reviewer comment 3: Authors should better describe the potential limitation of the study but also of the method (PET imaging for hypoxia) and possible future perspective with the use of more sophisticated approaches or by improving PET imaging. A paragraph in this sense before the conclusion is warmly requested.

Response 3: Thank you for your valuable suggestion. I have now included a dedicated paragraph before the conclusion discussing the potential limitations of both the study and the PET imaging method for hypoxia detection. This section addresses the challenges associated with PET imaging, such as spatial resolution limitations, radiotracer specificity, and variability in hypoxia assessment. Additionally, I have highlighted possible future perspectives, including advancements in PET imaging technology, the integration of multi-modal imaging, and the development of more sophisticated hypoxia-targeted tracers. I appreciate your feedback and welcome any further suggestions for improvement. The added section is below (Line 967-993):

“6. Limitations and future perspective

Despite the growing utility of PET imaging in hypoxia detection, several limitations must be acknowledged. One major challenge is the variability in tracer uptake, which can be influenced by factors such as blood flow heterogeneity, tumor microenvironment dynamics, and radiotracer pharmacokinetics. The spatial resolution of PET, though adequate for many clinical applications, is still limited when assessing hypoxia at a microscopic level, particularly in tumors with highly heterogeneous oxygenation. Additionally, the reliance on indirect hypoxia biomarkers, such as nitroimidazole-based tracers, introduces potential confounders, as their accumulation can be affected by factors beyond hypoxia, including redox status and cellular metabolism. Standardization of imaging protocols remains a critical issue, as differences in acquisition parameters, quantification methods (SUV, TBR, HF), and patient-specific physiological variations can impact reproducibility and inter-study comparisons.

To address these challenges, future advancements in PET imaging should focus on improving spatial resolution and sensitivity through novel detector technologies and improved reconstruction algorithms. Hybrid imaging modalities, such as PET/MRI, offer a promising avenue by combining PET's functional imaging capabilities with MRI’s superior soft-tissue contrast and oxygenation quantification. The development of next-generation hypoxia tracers with enhanced specificity, faster clearance, and improved target-to-background ratios is also essential. Additionally, artificial intelligence (AI) and machine learning algorithms could play a pivotal role in optimizing image analysis, automating tumor segmentation, and enhancing quantitative hypoxia assessment. The integration of PET with molecular and genetic profiling may also enable a more comprehensive understanding of tumor hypoxia, paving the way for personalized treatment strategies. Ultimately, continued research and technical advancements will be key to refining PET-based hypoxia imaging and expanding its clinical applications in oncology and beyond.”

Reviewer comment 4: Figure 9 should be enlarged to improve readability.

Response 4: Thank you for your feedback. I have now enlarged Figure 9 (which is now Figure 10 due to the addition of an extra figure as per another reviewer’s suggestion) to improve readability. Please let me know if any further adjustments are needed (line 761).

English enhancement: I have revised the manuscript to enhance the clarity and readability of the manuscript by improving sentence structure, grammar, and overall fluency. Technical terminology has been refined for precision, and complex sentences have been adjusted for better readability. Please let me know if any further linguistic improvements are needed. Please check highlighted changes in the track changes word file.

Reviewer 2 Report

Comments and Suggestions for Authors

the study seems to me well concerned even if I noted some issues that have to be solved to get the paper most readeble.

In the abstract at lines 20-21 the authors stated "... Additionally, 8% (15 
articles) explored other relevant topics, and 2% (4 articles) were unrelated to tumor 
hypoxia ...": in my opinion must be specified what are the "other relevant topics" and the "unrelated to tumor".

At line 164 the authors stated: "From the PET search, 189 articles were selected for abstract review". It is not clear, given the terms of research, which were the inclusion criteria and altgough the exlusion ones. In my opinion these criteria must be better specified in the materials and methods section.

At lines 169-173 the authors stated: "A substantial portion (39.7%, n = 
75) specifically focused on tumor hypoxia within lung malignancies, including non-small 
cell lung cancer (NSCLC) and other pulmonary cancers, highlighting its prognostic 
significance and role in treatment outcomes. Additionally, 8% (n = 15) of the references 
explored tumor hypoxia in other specific contexts or cancer type". Because the authors speak about "other pulmonary cancers" both referring to the 75 and 15 references they examined ("other specific contexts or cancer type") they have to specifiy if among the "other cancer" included in their 39,7% of the references are different from the "other specific contexts or cancer type" they included in the 8% of the references and which is the difference if it is.

The chapter 4.2 on "Molecular Mechanisms of Cancer Resistance to Radiotherapy and Chemotherapy" although interesting from a scientific point of view, in my opinion is out from the focus of the study and in its present form could get the paper less readble. I suggest to summarize this section of the paper.

The chapter "4.3. Hypoxia PET Imaging" in my opinion is too detailed when compared with the scope and the title of the paper: particularly I think that to treat and explain individually all the tracers could be the paper redundant. I note that the authors stated in the title that the paper in fon "non specialsit": in my opinion to write so deeply of every single tracer it is for "ultraspecialists", not fo non specialsit. I think that to summarize the main characteristics of the tracers in a table could get the paper more readible.

The conclusions in my opinion should be better specified in agree with the aims of the study as declared in the introduction because in the actual form they seem to short compared with the mole of the studies the authors examined

Author Response

Response to Reviewer 2

We sincerely appreciate your time and thoughtful feedback. Your insights have greatly helped refine our manuscript. We have carefully considered each suggestion and made revisions to improve clarity, accuracy, and presentation. Below, we provide detailed responses and outline the modifications made. Please note that cited line numbers refer to the track changes version (author-coverletter-44849621.v1.pdf)

Reviewer comment 1: In the abstract at lines 20-21 the authors stated "... Additionally, 8% (15 articles) explored other relevant topics, and 2% (4 articles) were unrelated to tumor hypoxia ...": in my opinion must be specified what are the "other relevant topics" and the "unrelated to tumor".

Response 1: Thank you for your valuable comment. We agree that our abstract findings were a bit vague. Therefore we enhanced and rewrote it to read as follows (Line 12-33):

“Abstract: Background: Hypoxia plays a critical role in lung cancer progression and treatment resistance by contributing to aggressive tumor behavior and poor therapeutic response. Molecular imaging, particularly positron emission tomography (PET), has become an essential tool for noninvasive hypoxia detection, providing valuable insights into tumor biology and aiding in personalized treatment strategies. Objective: This narrative review explores recent advancements in PET imaging for detecting hypoxia in lung cancer, with a focus on the development, characteristics, and clinical applications of various radiotracers. Findings: Numerous PET-based hypoxia radiotracers have been investigated, each with distinct pharmacokinetics and imaging capabilities. Established tracers such as ¹⁸F-Fluoromisonidazole (¹⁸F-FMISO) remain widely used, while newer alternatives like ¹⁸F-Fluoroazomycin Arabinoside (¹⁸F-FAZA) and ¹⁸F-Flortanidazole (¹⁸F-HX4) demonstrate improved clearance and image contrast. Additionally, ⁶⁴Cu-ATSM has gained attention for its rapid tumor uptake and hypoxia selectivity. The integration of PET with hybrid imaging modalities, such as PET/CT and PET/MRI, enhances spatial resolution and functional interpretation, making hypoxia imaging a promising approach for guiding radiotherapy, chemotherapy, and targeted therapies. Conclusion: PET imaging of hypoxia offers significant potential in lung cancer diagnosis, treatment planning, and therapeutic response assessment. However, challenges remain, including tracer specificity, quantification variability, and standardization of imaging protocols. Future research should focus on developing next-generation radiotracers with enhanced specificity, optimizing imaging methodologies, and leveraging multimodal approaches to improve clinical utility and patient outcomes”

Reviewer comment 2: At line 164 the authors stated: "From the PET search, 189 articles were selected for abstract review". It is not clear, given the terms of research, which were the inclusion criteria and altgough the exlusion ones. In my opinion these criteria must be better specified in the materials and methods section.

Response 2: Thank you for your valuable comment. We acknowledge that the previous section lacked clarity regarding the inclusion and exclusion criteria, which may have led to confusion, making it appear as if this were a strictly systematic review. To address this, we have removed the Materials and Methods section and replaced it with a revised section that clearly outlines the scope of our narrative review, ensuring a more transparent and structured presentation of the selected literature. This is the added section (Line 193-224):

“3. Narrative survey

A broad literature search was conducted to explore the use of PET and MRI for assessing hypoxia in cancer. Relevant studies published between 1990 and 2025 were considered, with a focus on original research and comprehensive reviews discussing imaging modalities for hypoxia detection. Studies that did not specifically address hypoxia imaging in cancer, lacked methodological detail, or were limited to commentaries, editorials, or conference abstracts without full-text availability were not included in the discussion.

The literature search identified numerous studies (approximately 730 articles) related to PET, PET/CT, and PET/MR imaging. Articles were selected based on their relevance to the topic, emphasizing key findings and advancements in hypoxia imaging. The selection process was guided by expertise in nuclear medicine, radiopharmacy, and imaging sciences, ensuring a well-rounded perspective. Key studies that contributed significant insights into hypoxia detection were highlighted and discussed in the review.

A total of 189 research references related to PET-based hypoxia detection and imaging were analyzed for this narrative review. Studies were included if they focused on PET imaging of tumor hypoxia, discussed the use of radiopharmaceuticals for hypoxia detection, or provided insights into the clinical and preclinical applications of hypoxia imaging in oncology. Studies were excluded if they lacked a direct focus on PET-based hypoxia imaging, were limited to commentary or editorial pieces without original data, or primarily discussed imaging modalities unrelated to PET.

Among the analyzed studies, 50.3% (n = 95) addressed tumor hypoxia in a general context, covering aspects such as molecular mechanisms, imaging techniques, and the impact of hypoxia on cancer progression and treatment resistance. A substantial portion, 39.7% (n = 75), specifically examined tumor hypoxia in lung malignancies, including non-small cell lung cancer (NSCLC) and other pulmonary cancers, emphasizing its prognostic significance and role in treatment outcomes. Additionally, 8% (n = 15) explored hypoxia in other cancer types, such as head and neck cancers, breast cancer, and gliomas. A small fraction, 2% (n = 4), were identified as unrelated to tumor hypoxia and were excluded from further discussion.

Through this review, we identified nine radiopharmaceuticals commonly used to evaluate tumor hypoxia, which are summarized in Table 1.”

Reviewer comment 3: At lines 169-173 the authors stated: "A substantial portion (39.7%, n = 75) specifically focused on tumor hypoxia within lung malignancies, including non-small cell lung cancer (NSCLC) and other pulmonary cancers, highlighting its prognostic significance and role in treatment outcomes. Additionally, 8% (n = 15) of the references explored tumor hypoxia in other specific contexts or cancer type". Because the authors speak about "other pulmonary cancers" both referring to the 75 and 15 references they examined ("other specific contexts or cancer type") they have to specifiy if among the "other cancer" included in their 39,7% of the references are different from the "other specific contexts or cancer type" they included in the 8% of the references and which is the difference if it is.

Response 3: Thank you for your insightful comment. We recognize the need for greater clarity in distinguishing the references categorized under lung malignancies (39.7%) and those classified under other specific cancer types (8%). To address this, we have revised the section to explicitly state that the 39.7% includes studies focusing on non-small cell lung cancer (NSCLC) and other primary lung malignancies, such as small cell lung cancer (SCLC). Meanwhile, the 8% category consists of studies on hypoxia in non-pulmonary cancers, including head and neck cancers, breast cancer, and gliomas. This distinction has been made clearer in the revised manuscript to avoid any confusion. It now reads as follows (line 214-224):

“Among the analyzed studies, 50.3% (n = 95) addressed tumor hypoxia in a general context, covering aspects such as molecular mechanisms, imaging techniques, and the impact of hypoxia on cancer progression and treatment resistance. A substantial portion, 39.7% (n = 75), specifically examined tumor hypoxia in lung malignancies, including non-small cell lung cancer (NSCLC) and small cell lung cancer , emphasizing its prognostic significance and role in treatment outcomes. Additionally, 8% (n = 15) explored hypoxia in other cancer types, such as head and neck cancers, breast cancer, and gliomas. A small fraction, 2% (n = 4), were identified as unrelated to tumor hypoxia and were excluded from further discussion.”

Reviewer comment 4: The chapter 4.2 on "Molecular Mechanisms of Cancer Resistance to Radiotherapy and Chemotherapy" although interesting from a scientific point of view, in my opinion is out from the focus of the study and in its present form could get the paper less readble. I suggest to summarize this section of the paper.

Response 4: Thank you for this valuable insight. We summerised this section and it now reads as follows (the text was reduced by 45%) (line 275-296):

Radiation therapy (RT) is a common treatment for solid tumors, often used alongside chemotherapy, immunotherapy, or surgery. RT damages DNA through ionizing radiation, causing direct damage when radiation is absorbed by DNA itself and indirect damage through radiolysis of surrounding water molecules, leading to the formation of reactive oxygen species (ROS) [11, 14, 38]. The presence of oxygen enhances ROS formation, increasing cellular damage, while hypoxia reduces this effect, leading to radiotherapy resistance [39]. Tumor cells attempt to repair radiation-induced DNA damage through mismatch repair, base excision repair, nucleotide excision repair, and double-strand break (DSB) repair, primarily via non-homologous end joining (NHEJ) and homologous recombination [14, 39]. However, due to dysfunctional DNA repair and hypoxia, cancer cells may survive radiation treatment, reducing its effectiveness [14, 39].

Hypoxia also influences gene expression, particularly through hypoxia-inducible factor (HIF-1α), which upregulates genes involved in angiogenesis and apoptosis (e.g., VEGF-1 and p53) [40, 41]. Additionally, HIF-1α is associated with chemotherapy resistance, further complicating cancer treatment [31]. Given that hypoxia is a negative prognostic marker, precise and efficient detection methods are crucial. Traditional monitoring techniques, such as oxygen polarographic needle electrodes and immunohistochemical analysis, are invasive [31, 42]. PET imaging has emerged as a promising non-invasive tool for real-time hypoxia monitoring by radiolabeling specific targets and receptors, allowing visualization of tumor metabolism and oxygenation levels in vivo [43].This highlights the critical role of PET imaging in addressing hypoxia-related treatment challenges and improving therapeutic outcomes.”

Reviewer comment 5: The chapter "4.3. Hypoxia PET Imaging" in my opinion is too detailed when compared with the scope and the title of the paper: particularly I think that to treat and explain individually all the tracers could be the paper redundant. I note that the authors stated in the title that the paper in fon "non specialsit": in my opinion to write so deeply of every single tracer it is for "ultraspecialists", not fo non specialsit. I think that to summarize the main characteristics of the tracers in a table could get the paper more readible.

Response 5: Thank you for your valuable feedback. We recognize that the current title may not fully reflect the balance between accessibility for non-specialists and the technical depth of the content. To better align with the scope of the paper, we suggest modifying the title to ensure it is more inclusive for both general readers and specialists. We would appreciate any further suggestions you may have regarding the title adjustment. The new title is :

“Hypoxia Imaging in Lung Cancer: A PET-Based Narrative Review for Clinicians and Researchers”

Reviewer comment 6: The conclusions in my opinion should be better specified in agree with the aims of the study as declared in the introduction because in the actual form they seem to short compared with the mole of the studies the authors examined

Response 6: Thank you for this valuable comment. We rewrote the conclusion to be more consistent with the study flow. It now reads as follows (Lines 996-1021):

“7. Conclusions

The ability to accurately detect and quantify tumor hypoxia is critical for optimizing cancer treatment strategies. Improved hypoxia imaging can play a pivotal role in the selection of chemotherapeutic agents that effectively target hypoxic tumor cells, as well as in enhancing the effectiveness of radiation therapy, which relies on oxygen availability for maximum therapeutic impact. A key consideration in hypoxia imaging is the distinction between acute and chronic hypoxia. While acute hypoxia can arise from transient changes in blood flow, chronic hypoxia is more relevant to tumor progression and treatment resistance. The focus of imaging should therefore be directed toward chronic hypoxia, which is associated with sustained metabolic and genetic adaptations that contribute to therapeutic resistance. Biomarkers associated with hypoxia, such as HIF-1α and others, provide valuable insight into these adaptations and should be considered in hypoxia-targeted imaging and treatment planning.

To further enhance the accuracy and clinical applicability of hypoxia PET imaging, standardization of imaging protocols and mathematical modeling are essential. Current challenges include variability in tracer kinetics, differences in PET acquisition and analysis methods, and the need for robust hypoxia thresholds. Addressing these challenges requires international standardization efforts (e.g., Image biomarkers standardization initiative), and this should be thoroughly discussed in  conferences, training workshops, and continuous updates in nuclear medicine research. Collaborative efforts between oncologists, radiopharmacists, and imaging specialists will be necessary to refine PET imaging techniques and expand their role in personalized cancer therapy.

Moving forward, the integration of advanced imaging modalities, improved radiotracers, and AI-based analytical tools will be key to overcoming current limitations in hypoxia imaging. These advancements will not only enhance diagnostic accuracy but also support precision medicine approaches, ultimately leading to better patient outcomes in lung cancer and other malignancies.”

English enhancement: I have revised the manuscript to enhance the clarity and readability of the manuscript by improving sentence structure, grammar, and overall fluency. Technical terminology has been refined for precision, and complex sentences have been adjusted for better readability. Please let me know if any further linguistic improvements are needed. Please check highlighted changes in the track changes word file.

Round 2

Reviewer 1 Report

Comments and Suggestions for Authors

Authors nicely addressed my concerns. Accordingly, I recommend the publication of the manuscript.

Reviewer 2 Report

Comments and Suggestions for Authors

The authors satisfied my suggestion and the paper in my opinion is now more readible